# FEW AND FEWER: LEARNING BETTER FROM FEW EXAMPLES USING FEWER BASE CLASSES

## ABSTRACT

When training data is scarce, it is common to make use of a feature extractor that has been pre-trained on a large "base" dataset, either by fine-tuning its parameters on the "target" dataset or by directly adopting its representation as features for a simple classifier. Fine-tuning is ineffective for few-shot learning, since the target dataset contains only a handful of examples. However, directly adopting the features without fine-tuning relies on the base and target distributions being similar enough that these features achieve separability and generalization. This paper investigates whether better features for the target dataset can be obtained by training on *fewer* base classes, seeking to identify a more useful base dataset for a given task. We consider cross-domain few-shot image classification in eight different domains from Meta-Dataset and entertain multiple real-world settings (domain-informed, task-informed and uninformed) where progressively less detail is known about the target task. To our knowledge, this is the first demonstration that fine-tuning on a subset of carefully selected base classes can significantly improve few-shot learning. Our contributions are *simple* and *intuitive* methods that can be implemented in any few-shot solution. We also give insights into the conditions in which these solutions are likely to provide a boost in accuracy. We release the code to reproduce all experiments from this paper on GitHub. https://anonymous.4open.science/r/Few-and-Fewer-C978

## 1 INTRODUCTION

Few-shot learning considers problems where training data is severely limited. It represents a challenge for deep learning, which typically requires large datasets of training examples (Wang et al., 2020b). The standard technique to leverage deep learning within few-shot tasks is to adopt some form of transfer learning, using a large distinct "base dataset" to train a model, which then serves as a feature extractor to integrate additional knowledge within the "target dataset", on which the task has to be solved. One of the most straightforward transfer strategies is thus to embed the target data into an appropriate feature space, and then to learn a simple classifier with minimal parameters in order to avoid overfitting to the few labeled examples (Wang et al., 2019).

However, the effectiveness of the transfer depends on the similarity of the base and target domains, and recent research in transfer learning (Oh et al., 2022) suggests that it may even have a deleterious effect if the domain gap is too large (Guo et al., 2020). This paper therefore considers the question: *can one reduce the domain gap by fine-tuning on the base classes that are the most similar to the target distribution?*. This approach aims to minimize the domain gap by concentrating the model's learning on a narrower, more relevant subset of the base classes closely aligned with the target distribution.

This questions the existence of universal feature extractors that would lead to *systematically* high performance on *any* few-shot task, a common trend in the field (Kirillov et al., 2023). Indeed, the growing body of literature on foundational models suggests that the optimal strategy for a new problem with limited data is to build on a foundational model which was trained on an Internet-scale dataset. Instead, in our approach, we show that tailored models can outperform generic ones on specific tasks, embodying the celebrated No Free Lunch theorem (Wolpert & Macready, 1997).

In this paper, we investigate a simple idea: given an off-the-shelf model trained on a base dataset – that we will call "base model", or "feature extractor" – we propose to fine-tune it using *only* the most

relevant classes from that same base dataset. By doing so, we aim to lower the importance of classes that could harm performance on the target task, while keeping a large enough pool of training data examples to ensure fine-tuning does not overfit.

Given a few-shot task and base dataset, we investigate the challenge of selecting a subset of base classes that, when used to fine-tune the feature extractor, leads to a feature representation with better inductive bias for the few-shot learning task. We consider eight target domains of Meta-Dataset (Triantafillou et al., 2019) in the cross-domain setting. We demonstrate that, for most but not all of the eight target domains, it is possible to obtain better target features for a Nearest Class Mean (NCM) classifier by fine-tuning the feature extractor with a subset of base classes from ImageNet. We later evaluate our method in multiple settings: *Domain-Informed* (DI), *Task-Informed* (TI) and *Uninformed* (UI) where progressively fewer details are known about the target task.

The main contributions of this work are:

- We demonstrate that fine-tuning with a subset of base classes can improve accuracy.
- We present simple methods to select such a subset given varying degrees of information about the few-shot task (either the few-shot examples themselves or unlabelled examples from the target domain).
- We investigate the feasibility of employing a static library of feature extractors that are fine-tuned for different class subsets. We compare several methods for deciding these class subsets ahead of time, and several heuristics for identifying a useful class subset at runtime.

## 2 BACKGROUND AND RELATED WORK

**Terminology.** A few-shot classification task (or episode) comprises a support set for training the classifier and a query set for testing the classifier. The support set contains a small number of examples for each class. If we have $K$ classes with $N$ examples for each, then we refer to the problem as "$N$-shot $K$-way" classification. When benchmarking few-shot solutions, accuracy is measured on the query set, and averaged over a large number of different tasks. Depending on the application case, one may consider inductive few-shot learning, where each query is classified independently, or transductive few-shot learning, where all queries are processed jointly, meaning that the classifier can benefit from the added information coming from their joint distribution. In this paper, we focus on inductive few-shot learning, although the techniques could be extended to the transductive setting.

**Few-shot paradigms.** To solve a few-shot task, the main idea found in the literature is to rely on a pre-trained feature extractor, trained on a large generic dataset called the "base dataset". Several strategies on how to train efficient feature extractors have been proposed, including meta-learning methods (Finn et al., 2017) or closed-form learners (Snell et al., 2017; Bertinetto et al., 2018; Yoon et al., 2019), others directly learn a mapping from support examples and a query input to a prediction (Vinyals et al., 2016; Ravi & Larochelle, 2017; Garcia Satorras & Bruna Estrach, 2018; Requeima et al., 2019; Hou et al., 2019; Doersch et al., 2020). But simple straightforward classical batch learning of feature extractors have also shown to achieve State-Of-The-Art performance (Bendou et al., 2022b). This is why we rely on such simpler feature extractors in our work. Once a feature extractor is chosen, many adaptation strategies have been proposed (Wang et al., 2019; Triantafillou et al., 2019). Simple classifiers such as Nearest Neighbor or **Nearest Class Mean (NCM)** without additional learning (Wang et al., 2019; Bateni et al., 2020; Snell et al., 2017) have shown competitive performance, hence we adopt this approach for its simplicity and effectiveness. Based on recent evidence (Luo et al., 2023), we have strong reasons to believe that the proposed methodology could lead to improvements for any feature extractor training algorithm.

**Lightweight adaptation of feature extractors.** Several works have previously sought to obtain task-specific feature extractors for few-shot learning. This is typically achieved by introducing a small number of task-specific parameters into the model in the form of residual adapters (Rebuffi et al., 2017) or Feature-wise Linear Modulation (FiLM) layers (Perez et al., 2018). In the multi-domain setting, these parameters can be simply trained for each domain (Dvornik et al., 2020; Liu et al., 2021a). Otherwise, the task-specific parameters must either be trained on the support set (Li et al., 2022) or predicted from the support set via meta-learning (Bertinetto et al., 2016; Oreshkin et al., 2018; Requeima et al., 2019). While feature adaptation has proved effective for multi-domain

few-shot learning, it is difficult to apply to the cross-domain setting due to the need to train on the support set. This paper instead proposes to update *all* parameters of the feature extractor by re-visiting the base dataset and fine-tuning only on a subset of relevant classes.

**Selecting feature extractors or class subsets.** In our work, we consider a setting which requires selecting amongst feature extractors that were each fine-tuned on a subset of base classes. This requires predicting the downstream performance of a feature extractor, a question that has previously been considered by Garrido et al. (2022). They proposed the RankMe metric based on a smooth measure of the matrix rank (Roy & Vetterli, 2007). Achille et al. (2019) considered the problem of measuring task similarity using the Fisher Information Matrix (FIM), and demonstrated the ability of their proposed metric to select a feature extractor trained on an appropriate subset of classes. The experimental section will show that straightforward measures such as cross-validation error perform at least as well as these more involved measures when using a simple classifier in the few-shot setting. Dvornik et al. (2020) used a linear combination of features from domain-specific feature extractors, with coefficients optimized on the support set. We do not consider mixtures of features in this paper, as our goal is not to obtain the highest accuracy but rather to investigate whether accuracy can be improved using *fewer* base classes.

**Re-using the base dataset in transfer learning.** The prevention of over-fitting is critical when seeking to train with small datasets. Whereas several works have considered regularization strategies such as selective parameter updates (Shen et al., 2021) or auxiliary losses (Su et al., 2020; Majumder et al., 2021; Wang et al., 2020a; Islam et al., 2021), our strategy is to re-use a subset of the base dataset given knowledge of the downstream task. This high-level idea is not novel in itself outside the context of few-shot learning, as several works have considered ways to use the base dataset beyond task-agnostic pre-training. Liu et al. (2021b) showed that transfer learning could be improved by retaining a subset of classes from the base dataset during fine-tuning, using separate classifiers and losses for the examples from different datasets. Besides manual selection of classes, they proposed to obtain class subsets by solving Unbalanced Optimal Transport (UOT) for the distance between class centroids in feature space. Earlier works used low-level image distance to select a subset of examples (rather than classes) to include in fine-tuning (Ge & Yu, 2017) or instead selected a subset of classes at the pre-training stage before fine-tuning solely on the target dataset (Cui et al., 2018). While the works which select class subsets are the most closely related to this paper, all rely on fine-tuning on the target set and do not consider the few-shot regime, entertaining at minimum about 600 examples (corresponding to 20% of Caltech 101). In contrast, this work will focus on few-shot learning where it is difficult to fine-tune on the support set (Luo et al., 2023). We consider subsets of classes rather than examples because it simplifies the problem, ensures that the dataset remains balanced, and provides an intuitive mechanism to select a feature extractor.

**Domain adaptation.** We also consider in our work a Domain-Informed (DI) setting, that bears some resemblance to methods based on domain adaptation (Sahoo et al., 2019; Khandelwal & Yushkevich, 2020) as it makes use of unsupervised data from the target domain.

## 3 FEATURE EXTRACTORS FOR FEWER BASE CLASSES

### 3.1 FORMULATION

The simple few-shot pipeline that we consider comprises three key stages:

Step 1. Train a feature extractor with parameters $\theta$ using a labeled base dataset with classes $\mathcal{C}$;
Step 2. Fine-tune the feature extractor on a subset of base classes $\mathcal{C}' \subset \mathcal{C}$ to obtain $\theta'$;
Step 3. Extract features for the query and support sets and perform NCM classification.

The feature extractor takes the form of a deep neural network $h_\theta : \mathbb{R}^m \to \mathbb{R}^n$. It is trained in combination with an affine output layer $g$ such that the composition $g \circ h_\theta$ minimizes the softmax cross-entropy loss (Bendou et al., 2022b). The NCM classifier simply computes the centroid of each class in feature space, and then classifies new examples according to minimum Euclidean distance.

The canonical approach to few-shot learning (Wang et al., 2019) jumps directly from Step 1 to Step 3, effectively setting $\theta' = \theta$. However, this makes strong demands of the feature extractor: it is expected to be *universal* in that its representation must be immediately applicable to any downstream task, even those belonging to different domains. Whilst many past works in few-shot learning have focused on improvements to Step 3, we consider the hypothesis that fine-tuning on *fewer* base classes

(and hence a smaller dataset) may in fact *improve* accuracy in few-shot tasks. A 2D visualization of the effect on a 3-way task is shown in the Appendix.

We now turn to the problem of identifying a suitable class subset $\mathcal{C}'$. We consider three different settings for class subset selection, which are defined by different degrees of knowledge of the task, and consider different constraints on running time. **Task Informed (TI)** selection considers the scenario where the support set $\mathcal{S}$ can itself be used to select the class subset $\mathcal{C}'$. This represents the ideal scenario, although the computational effort involved in fine-tuning (on a subset of the base dataset) may be prohibitive if many few-shot problems need to be solved, or if a classifier must be obtained quickly. **Domain Informed (DI)** selection considers the scenario where one cannot afford to fine-tune a feature extractor for each few-shot task, yet a dataset $\mathcal{D}$ comprising a superset of classes from the same domain as $\mathcal{S}$ is available for the purpose of class-subset selection (without requiring labels). This could correspond to a realistic scenario where a robot is exploring an environment, generating a large number of unlabeled images from the target domain. As the number of shots in the support set decreases, DI selection also has the advantage of giving a lower variance estimate of the class subset than TI, since it uses a larger set of examples. However, this comes at the cost of a higher bias, since the examples do not correspond exactly to the few-shot task. Finally, **Uninformed (UI)** selection considers the problem of defining multiple class subsets $\mathcal{C}'_1, \ldots, \mathcal{C}'_L$ ahead of time without knowledge of the target domain, and incurs the additional problem of having to select the most suitable class subset (and associated feature extractor) for a given support set. This setting is particularly interesting for applications where there are strong constraints in terms of computational effort or latency, seeking a general-purpose set of specialists.

The key baselines to consider will be the canonical approach with an NCM classifier (i.e. excluding Step 2 above), and **fine-tuning on the support set (S)**. The remainder of this section will address the design of techniques for selecting class subsets in each setting.

## 3.2 Choosing class subsets: Informed settings

The informed settings (TI, DI) consider the problem of selecting a subset of base classes $\mathcal{C}' \subset \mathcal{C}$ given a set of examples $\mathcal{X} = \{x_i\}_i$. In TI selection, $\mathcal{X}$ would be the support set, whereas in DI selection, $\mathcal{X}$ would be the domain examples $\mathcal{D}$ ignoring the labels. The class subset $\mathcal{C}'$ will then be used to fine-tune the "base" feature extractor, which was trained on the entire base dataset.

To choose a class subset, we need a method by which to identify the base classes which are most useful for a given this set of examples $\mathcal{X}$. Fortunately, the base model already comprises a classifier that assigns a score to each base class. We therefore propose to simply compute the average class likelihoods predicted by the base model on the novel set $\mathcal{X}$, and then select the $M$ highest-scoring classes. This straightforward selec-

---

**Algorithm 1** Average Activation selection (TI, DI)

---
**Require:** Base classes $\mathcal{C}$, examples $\mathcal{X} = \{x_i\}$, pre-trained model with feature extractor $h$ and classifier $g$, class subset size $M = 50$
1: Compute average scores
$\quad p = \frac{1}{|\mathcal{X}|} \sum_{x_i \in \mathcal{X}} \text{softmax}(g(h(x_i)))$
2: Sort $p$ in descending order
3: **return** $\mathcal{C}' :=$ First $M$ classes of $p$

---

tion strategy will henceforth be referred to as **Average Activations (AA)**, and is outlined in Algorithm 1. While there is no guarantee that this procedure will select the class subset that yields the optimal representation for the final task after fine-tuning, it is a cheap and reasonable proxy for that purpose. Note that we use $M = 50$ in all experiments to have subset sizes comparable to the UI setting, described in the following section.

As a point of reference, we also consider a more sophisticated selection strategy that requires labels for the set of examples $\mathcal{X}$ that informs selection. We adopt the Unbalanced Optimal Transport (UOT) formulation of Liu et al. (2021b), which assigns unit mass to the classes in $\mathcal{X}$ and $\mathcal{C}$, and uses the distance between class centroids to define the cost matrix. All regularization parameters are set as in (Liu et al., 2021b), and we similarly take the top $M = 50$ base classes according to the resulting (unnormalized) marginals on $\mathcal{C}$.

## 3.3 Choosing class subsets: Uninformed setting

The uninformed setting considers the case where it is infeasible to fine-tune the model on demand. Our aim is thus, with *off-the self* tools, to construct a *static library* of specialist feature extractors

from class subsets that are determined in an unsupervised manner, such that a suitable class subset can then be chosen in light of the support set. To this end, we perform agglomerative hierarchical clustering of the base classes using Ward's method (Ward Jr, 1963), where each class is represented using either its centroid under the base feature extractor $h_\theta$ (visual features, V) or a vector embedding of its name from the text encoder of the publicly available CLIP model (Radford et al., 2021) (semantic features, Se.). Final clusters were obtained by choosing a threshold on distance that gave a total of eleven relatively balanced clusters for the 712 classes in the ImageNet training split of Meta-Dataset (Triantafillou et al., 2019). The same process was performed for the concatenation of visual and semantic features (denoted X), with the two types of feature vectors being normalized and centered prior to concatenation. To obtain a comparable baseline for the clustering process, we further construct a random (R) partitioning of the base classes into eleven subsets. Following clustering, a different feature extractor is independently fine-tuned for each class subset, yielding a static library of class subsets and model parameters $(\mathcal{C}'_j, \theta'_j)$. The base model $(\mathcal{C}, \theta)$ is also included in the static library.

### 3.4 Heuristics for selecting a feature extractor

Lastly, we turn to the problem of selecting between specialist feature extractors given the support set for a novel few-shot task. For this purpose, we consider a collection of heuristics that are expected to correlate with accuracy on the query set. Heuristics can make use of the labeled support set $\mathcal{S}$, the feature extractor $h_{\theta'_j}$ and the class subset $\mathcal{C}'_j$ which was used for fine-tuning.

We briefly describe the heuristics here, please refer to the Appendix for a more complete description. The most obvious heuristics to include are the accuracy and maximum confidence on the support set (SSA and SSC, respectively) and the leave-one-out cross-validation accuracy (LOO). We also consider the Signal to Noise Ratio (SNR) defined by the comparison of within-class and between-class covariances. We incorporate RankMe (RKM), Monte-Carlo Sampling (MCS) and Fisher Information Matrix (FIM) metrics from past work: RankMe (Garrido et al., 2022) considers the (smooth) rank of the feature matrix with the motivation that good features should exhibit linear independence, Monte-Carlo Sampling (Bendou et al., 2022a) obtains virtual examples by sampling from regularized Gaussian distributions that have been fit to each class in the support set to construct an artificial validation set, and the Fisher Information Matrix (Achille et al., 2019) provides a measure of task similarity using a probe network. Finally, while Average Activation (AA) was previously used as a subset selection method, our use of class subsets to define the feature extractors enables it to be employed as a heuristic. This is achieved by selecting the class subset which has the greatest cumulative activation in the support set of a task.

With the notable exception of AA and SNR, all heuristics are inapplicable to the one-shot setting, since they require at least two examples per class to construct a validation set or measure within-class covariance. SNR circumvents this issue by considering only between-class covariance in the one-shot setting. Further note that, besides AA, all heuristics involve evaluation of the candidate feature extractor, hence selecting a class subset will involve exhaustive evaluation of all feature extractors in the library, which is typically only on the order of tens of models.

To validate the effectiveness of our heuristics, we compare them to a random heuristic (RH) which selects a feature extractor uniformly at random and to an oracle which always selects the feature extractor with the highest accuracy on the validation set. The latter reveals an upper bound on the best possible performance for a given set of few-shot tasks and feature extractors. Its performance might not be achievable given the information at hand.

## 4 Experiments

We report results on the eight datasets within Meta-Dataset excluding ImageNet and QuickDraw. These include Omniglot (handwritten characters), Aircraft, CUB (birds), DTD (textures), Fungi, VGG Flowers, Traffic Signs and MSCOCO (common objects) (Lake et al., 2015; Maji et al., 2013; Wah et al., 2011; Cimpoi et al., 2014; Schroeder & Cui, 2018; Nilsback & Zisserman, 2008; Houben et al., 2013; Lin et al., 2014). Recall that S denotes the approach of fine-tuning on the support set. We consider three sampling procedures for generating few-shot tasks: 1-shot 5-ways, 5-shots 5-ways, and the task-sampling procedure described by Meta-Dataset (Triantafillou et al., 2019), denoted

Table 1: Performance change using fine-tuning on the support (S), with a Task-Informed (TI) subset selection, a Domain-Informed (DI) subset selection, and DI-UOT subset selection. All positive boosts with overlapping confidence intervals are bolded. Overall DI performs the best followed by TI. S performs the worst. UOT selection strategy is outperformed by simple AA selection. The complete table with UOT on each dataset is in the appendix.

| Dataset | Method | 1-shot 5-ways | | 5-shots 5-ways | | MD | |
|---|---|---|---|---|---|---|---|
| | | Baseline | $\Delta$ | Baseline | $\Delta$ | Baseline | $\Delta$ |
| Aircraft | S | 39.95 ±0.70 | -3.60 ±0.64 | 63.18 ±0.74 | -1.48 ±0.61 | 65.86 ±0.90 | **+5.33 ±0.69** |
| | TI | | -0.06 ±0.33 | | **+0.26 ±0.31** | | +1.33 ±0.25 |
| | DI | | **+0.34 ±0.32** | | **+0.54 ±0.31** | | +1.32 ±0.27 |
| CUB | S | 64.34 ±0.90 | -19.28 ±0.88 | 87.78 ±0.59 | -18.97 ±0.63 | 79.29 ±0.90 | -14.51 ±0.60 |
| | TI | | +2.64 ±0.44 | | **+2.16 ±0.26** | | +1.08 ±0.19 |
| | DI | | **+3.27 ±0.44** | | **+2.29 ±0.26** | | **+2.20 ±0.20** |
| DTD | S | 45.21 ±0.77 | +0.66 ±0.77 | 70.10 ±0.59 | -3.12 ±0.59 | 76.03 ±0.69 | -6.67 ±0.69 |
| | TI | | **+2.85 ±0.46** | | **+2.77 ±0.33** | | **+2.44 ±0.29** |
| | DI | | **+2.90 ±0.48** | | **+2.96 ±0.33** | | **+2.78 ±0.31** |
| Fungi | S | 53.01 ±0.92 | -6.59 ±0.74 | 74.87 ±0.80 | -8.33 ±0.62 | 51.57 ±1.16 | -15.05 ±0.53 |
| | TI | | **+0.92 ±0.39** | | **+1.67 ±0.30** | | **+1.07 ±0.26** |
| | DI | | **+1.07 ±0.41** | | **+1.89 ±0.29** | | **+1.38 ±0.25** |
| Omniglot | S | 61.80 ±1.03 | -3.16 ±1.11 | 81.53 ±0.76 | **+3.53 ±0.85** | 59.51 ±1.31 | -4.59 ±1.07 |
| | TI | | **+2.65 ±0.38** | | **+2.94 ±0.29** | | **+3.74 ±0.23** |
| | DI | | **+3.52 ±1.22** | | **+3.57 ±0.81** | | **+3.93 ±0.61** |
| MSCOCO | S | 43.91 ±0.85 | -5.44 ±0.66 | 63.04 ±0.79 | -6.20 ±0.63 | 44.99 ±0.99 | -17.00 ±0.72 |
| | TI | | **+1.27 ±0.35** | | **+1.87 ±0.29** | | +1.85 ±0.17 |
| | DI | | **+1.62 ±0.34** | | **+2.09 ±0.30** | | **+2.25 ±0.17** |
| Traffic Signs | S | **57.35 ±0.85** | -4.67 ±0.66 | 74.11 ±0.78 | **+6.17 ±0.62** | **53.77 ±1.05** | **+0.77 ±1.00** |
| | TI | | -0.84 ±0.32 | | -1.22 ±0.25 | | -2.02 ±0.17 |
| | DI | | -0.79 ±0.95 | | -1.48 ±0.77 | | -1.82 ±0.44 |
| VGG Flower | S | 75.86 ±0.84 | +0.19 ±0.79 | 94.46 ±0.33 | -1.45 ±0.37 | 92.77 ±0.58 | -5.18 ±0.51 |
| | TI | | **+2.04 ±0.40** | | **+0.64 ±0.18** | | **+1.03 ±0.16** |
| | DI | | **+1.88 ±0.41** | | **+0.52 ±0.18** | | **+0.84 ±0.16** |
| **Average** | S | | -5.24 ±0.78 | | -3.73 ±0.61 | | -7.11 ±0.73 |
| | TI | | **+1.43 ±0.38** | | **+1.39 ±0.28** | | **+1.31 ±0.21** |
| | DI | | **+1.73 ±0.57** | | **+1.55 ±0.41** | | **+1.61 ±0.30** |
| | DI-UOT | | +0.63 ±0.47 | | +0.36 ±0.33 | | +0.32 ±0.28 |
| | TI-UOT | | **+1.43 ±0.36** | | **+1.10 ±0.44** | | **+1.21 ±0.32** |

MD, whose tasks have a much larger but varying number of shots and ways. We report the baseline accuracy and the change in performance with respect to the baseline or *boost*, denoted $\Delta$. A fixed set of 600 few-shot tasks is sampled for each dataset and sampling procedure, and this is held constant for all methods (S, TI, DI, DI-UOT, TI-UOT). Since accuracy is measured using the same set of tasks for all methods, the confidence interval of the accuracy boost can be computed using paired trials. The confidence intervals for the baselines instead represent the distribution of the sample mean across the 600 different tasks.

## 4.1 EFFECT OF INFORMED CLASS SELECTION

Our first main experiment investigates the change in accuracy effected by fine-tuning the feature extractors on a subset of base classes before performing NCM classification, considering the Average Activation selection strategy in both the Task-Informed and Domain-Informed settings. This is compared to the effect of fine-tuning on the support set, as well as the UOT selection strategy (Liu et al., 2021b) in DI and TI. Table 8 reports baseline accuracies and relative boosts in all settings for each dataset and few-shot sampling procedure.

The results reveal that Domain-Informed selection of base classes can significantly improve accuracy. The average boost across all datasets and samplings using DI selection is $+1.62 \pm 0.08$ points. Examining individual datasets, we note the consistent negative change in accuracy on Traffic Signs, with the exception of fine-tuning given a minimum number of shots. This is likely explained by the absence of similar images in ImageNet. Indeed, whereas the ImageNet activations for CUB are

distributed across roughly 50 bird classes, the most strongly activated class for Traffic Signs is *Nematode*, far outside the domain of traffic signs. Poor improvements are observed on Aircraft, since ImageNet contains only few relevant classes (*airliner* and *military plane*) which are likely supersets of the classes in the few-shot task. These results explain the large variability in boost achieved in the DI setting, and are detailed in the Appendix.

One hypothesis which is not borne out in the experimental results is that class selection can only achieve significant improvements for tasks which are relatively easy, or where the base feature extractor is already relatively effective. If anything, the boost tends to be inversely correlated with the accuracy, with larger improvements being achieved when the accuracy of the baseline is lower (as shown in the Appendix). Another hypothesis which will require further investigation is that Aircraft and Traffic Signs perform poorly because they require the feature extractor to represent shape more than color or high-frequency texture, whereas these are useful cues for datasets such as CUB, VGG Flower and DTD.

From the results, we observe the strategy based on Unbalanced Optimal Transport (Liu et al., 2021b) to achieve improvements that are only on-par or worse than the naive Average Activation strategy. In particular, we observe a large drop in performance on Omniglot, whose test split contains the largest number of classes (659), revealing that the hyperparameters of the algorithm are likely sensitive to the size of the problem. The set of classes selected using UOT varies significantly from that selected using AA; we observed that the Intersection over Union of these sets ranged between 22% for MSCOCO and 78% for CUB.

Task-Informed selection is often observed to somewhat under-perform Domain-Informed selection. This is particularly pronounced in CUB, for which the base dataset contains a large number of relevant classes (birds) which could be retrieved for the class subset. This observation points to the higher variance of selecting class subsets from fewer examples (as shown in the Appendix). This suggests that the bias of Domain-Informed selection is preferable to the variance of Task-Informed selection, which remains true even in higher data regimes.

Fine-tuning on the support set (S) can be rewarding, especially in the higher data regimes of 5-way 5-shot and MD task sampling, where boosts of up to $\Delta = +6.17 \pm 0.62$ points are achieved for 5-way 5-shot classification on Traffic Signs. We note that the accuracy of the baseline is particularly low on Traffic Signs, probably due to the lack of relevant data in the base dataset. In this case, fine-tuning on the support set is likely to have a large positive effect where other methods can only amplify or attenuate the influence of relatively unrelated classes in the base dataset. The same phenomenon may also be at play on a smaller scale for Aircraft. During experimentation, we observed that fine-tuning on the support set is particularly sensitive to the choice of hyperparameters. Amongst all configurations we tested (see Appendix for details), fine-tuning on the support set typically led to a significant decrease of performance. We advocate that finding the right hyperparameters for each task without a validation set is the real bottleneck for this method.

When the domain is known, Domain-Informed selection is the most reliable approach to increase few-shot accuracy. This is especially the case for the low data regime of 1-shot 5-ways, as it greatly benefits from the information contained in the unlabeled examples. In a mixed sampling where more shots are available, DI still retains its advantage, although the gap is reduced. When the Domain is unknown, Task-Informed selection remains a safer option than fine-tuning on the support set, which can have a catastrophic outcome.

Overall, the table clearly shows that training with fewer base classes can indeed lead to significant boosts in accuracy compared to the feature extractor, supporting the claim that fine-tuning with a subset of base classes can improve accuracy. What is more, we measured this increased separability using the silhouette score (Rousseeuw, 1987). Across all datasets, the silhouette score of target features increased by $\Delta = +0.0103$ with an average silhouette score for the baseline of -0.001.

## 4.2 UNINFORMED SETTING

Our second main experiment considers the Uninformed (UI) setting. Specifically, we seek to determine whether a positive boost relative to the baseline can be achieved without knowledge of the task during fine-tuning, and compare methods for the unsupervised construction of class subsets as well as the selection of feature extractors. The results are reported in Figure 1, which presents

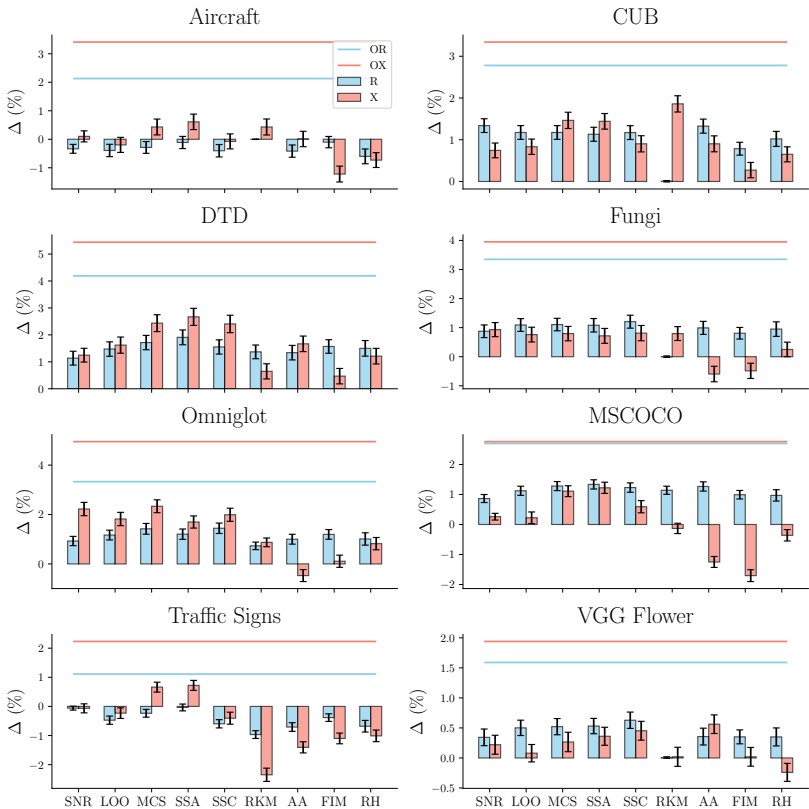

Figure 1: Difference of accuracy with baseline after feature extractor selection using heuristics. Tasks are sampled following the MD protocol. In R (resp. X) heuristics select a feature extractor amongst the R (resp. X) library of feature extractors. The oracle OR (resp. OX) selects the best feature extractor for each task in the R (resp. X) library. The Random Heuristic (RH) picks a random feature extractor. SSA and MCS are the two best performing heuristics. A meaningful choice of class (X) is desirable in particular on datasets with high boosts.

the boost in performance for each domain and selection heuristic using MD sampling with both the concatenated (X) and random (R) subset constructions.

First, we point out that in most cases we obtained significant boost in accuracy. MCS and SSA consistently led to a positive impact across all our experiments, when combined with the X design of subsets. We consider this result important as it clearly outlines the ability to deploy such solutions in applications where strong constraints apply in terms of computations and/or latency. This experiment supports our second claim from the introduction.

It is not a surprise that X generally outperforms R in particular on datasets where improvements are large, showing that a meaningful design of subsets is preferable. We also note that the X-based oracle oftentimes reaches a much higher accuracy than its R-based counterparts. However, some heuristics such as like AA and FIM seem particularly detrimental to X. This does not occur for MSCOCO, a dataset of natural images which is quite close to the ImageNet distribution. This suggests that it is most important to use a meaningful construction of subsets when the target dataset is more fine-grained or less similar compared to the base dataset. Results for V, Se. (in the Appendix) and X are on-par with a slight advantage for V particularly on the Traffic Signs dataset. We nonetheless preferred to present results for X as it combines two orthogonal cues and is therefore likely to be more robust in novel domains.

Finally, amongst the different heuristics, Support Set Accuracy (SSA) performs the best under MD sampling on average across datasets and subset constructions, with an average boost of $\Delta = +1.13 \pm 0.22$ points. For 5-shot 5-way tasks, Monte-Carlo Sampling (MCS) is the best with a boost of $\Delta = +0.78 \pm 0.27$ points, while in 1-shot 5-way tasks, the Signal to Noise Ratio (SNR) heuristic yields the best boost with $\Delta = +0.74 \pm 0.38$ points. Thus, interestingly, even in the adversarial conditions of a single shot per class, it is still possible to expect a significant boost in accuracy by

adopting a feature extractor which is fine-tuned for a pre-determined subset of base classes. The large gap to the oracle (denoted by O) indicates that the maximum achievable boost is consistently above 2% and can range as high as 6%. Compared to previous work (FIM (Achille et al., 2019), RKM (Garrido et al., 2022)), our heuristic performs significantly better. The heuristic based on Average Activation of the base classifier was unfortunately found to be unreliable across domains when compared to heuristics which directly assess NCM classifier on the support set.

## 4.3 IMPLEMENTATION DETAILS

In TI and S, a fine-tuning is performed for each task. Therefore, we could not afford to explore the hyperparameter space for each case. In particular, in the TI setting where a complete two steps fine-tuning with 50 classes had to be performed for each task, each dataset and each sampling setting. Please note that in the DI setting we make use of the validation split to choose our class subsets so as to make it task independent while remaining domain dependant. We use an Adam (Kingma & Ba, 2014) optimizer to fit the classifier (first step) and SGD with a Nesterov momentum of 0.9 (Polyak, 1964) for the complete fine-tuning (second step). We used a learning rate of 0.001 and a cosine scheduler (Loshchilov & Hutter, 2016) in every setting for comparability. We also limit the dataset size to 10k examples in order to isolate the effect of the choice of data. We fine-tune on 10 epochs during the first step (frozen feature extractor) and 20 steps on the second step (unfrozen feature extractor). We use a simple ResNet-12 architecture as feature extractor. We show in the Appendix that DI can be improved by using heuristics to select between feature extractors fine-tuned with different learning rate. We use off-the-shelf standard procedures to train $f_\theta$, such as the ones described in Bertinetto et al. (2018); Bendou et al. (2022b). We used 2 clusters with GPUs. Namely, we used Nvidia A100s and V100s to run our experiments. A machine equipped with an Nvidia 3090 was used to prototype our methods.

## 4.4 DISCUSSION

We could also show that our results extend to segmentation task as shown in Table 5 in the appendix. Our work touches on a wide range of questions of which many could not be investigated in this work. In particular we only shallowly address the geometric and ontological relationships between the source and target classes. These relationships are probably key to predict the sign and magnitude of accuracy boost. We fixed the number of clusters in the UI setting and the number of selected classes in the DI and TI setting although we show in the appendix the effect of changing the number of selected classes. Future work could include an analysis of our methods in the context of a domain shift between the support and query examples (Bennequin et al., 2021).

Another limitation of our work is the high computational cost of some heuristics (FIM, MCS and LOO) and settings (TI, TI-UOT and to a lesser extent S). As mentioned earlier, fine-tuning on the support set can be very rewarding but often comes with difficulties to set good hyperparameters. As such, we think that methods aiming at predicting the accuracy of a few-shot task could be of tremendous interest to set them appropriately. Furthermore, self-supervised fine-tuning may prove to be a superior solution in certain circumstances. What is more, we believe that fine-tuning is not the *be-all and end-all* solution to adapt embeddings for a task. Carefully crafted, data-dependent, projections might be fast "on-the-fly" solutions to increase performances.

## 5 CONCLUSION

In conclusion, in this paper we introduced various ways to identify relevant subset of base classes that can, if fine-tuned on, significantly improve accuracy when facing a few-shot task. Interestingly, fine-tuning on a subset selected using the unlabelled target domain seems to be the most reliable way to improve performances. This however does not apply to all datasets, meaning that a lot of open questions remain. We hope that this inspires the community to investigate this effect, including the role of dataset scale. We also introduced a simple strategy of building an offline static library of feature extractors from which can be dynamically selected one when facing a few-shot task. With the rise of interests in foundational models that are candidates to be universal embedding for downstream tasks, we think our work can be an interesting opposing view for future research.

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

## A APPENDIX

### A.1 IMPACT OF LEARNING RATE ON FINE-TUNING (DI SELECTION)

We fine-tuned the baseline model on DI-selected subsets with varying learning rates. Figure 3 shows that the boost in accuracy compared to the baseline is highly dependent on the learning rate chosen.

Since the learning rate can have a strong effect on the final accuracy, we further propose to use our heuristics to determine which learning rate is the most suitable. Note that we cannot use AA nor FIM to select learning rates of feature extractors. These two methods purely on the choice of data independently of the model used. We observe strong gains in performance in Figures 4 and 5 on most datasets. These Figures show that for cetain datasets the learning rate of 0.001 that was reported in the main paper can be outperformed using the heuristics to select the best learning rate.

### A.2 ABOUT BATCH NORMALIZATION DURING FINE-TUNING

During the fine-tuning process, both the feature extractor parameters and batch normalization statistics can be adapted to the selected subset. The latest statistics are estimated using moving averages whereas parameters are updated through gradient descent. Thus, we consider experiments where the learning rate is set to 0, meaning that only the batch normalization statistics are updated. Figure 3 shows that Batch Normalization alone can be highly beneficial to the performance. This is particularly the case for Omniglot.

### A.3 A CLOSER LOOK AT THE UNSUPERVISED SELECTION OF CLASSES

We show the dendrograms built with Ward's method below. We take a closer look at the cluster of birds in 12 and 13. We show that both semantic and visual features demonstrate an impressive ability to identify a cluster related to birds. However both identified clusters contain anomalies (non-birds) and miss some birds. More details are in the captions. This is the rationale for our method of selection in using both visual and semantic features (X).

### A.4 DESCRIPTION OF HEURISTICS

(Minor correction to main text: In addition to the SNR and AA heuristics, the FIM heuristic can also be used with only a single shot.)

**Leave-One-Out (LOO).** LOO is a form of cross-validation on the support set. Validation is acquired by randomly sampling a single element from the support set, making it unused in the estimation of class centroids. The process is repeated a large number of times and averaged to obtain an estimated accuracy. To accelerate the calculations, we isolate one sample *per class* in our experiments. LOO is required as it impacts the training set as little as possible. In few-shot conditions, removing even just a few elements from the support set is expected to yield significant impact to the performance of the considered classifier.

**Signal-to-Noise Ratio (SNR).** SNR is another metric that relates to the accuracy. In an isotropic Gaussian case, it is a perfect heuristic for a theoretical accuracy. We define it as such for 2 classes $\{i, j\}$:

$$SNR(i,j) = \frac{\delta}{\xi} = 2\frac{\|\mathbb{E}(\mathcal{N}^i) - \mathbb{E}(\mathcal{N}^j)\|_2}{\sigma(\mathcal{N}^i) + \sigma(\mathcal{N}^j)} \tag{1}$$

where $\mathbb{E}(\mathcal{N}^i)$ and $\sigma(\mathcal{N}^i)$ are respectively the empirical expectation and standard deviation of the class $i$ in the support set, $\delta$ is the margin and $\xi$ is the noise. For more than two ways tasks we compute the average over all couple of classes.

**Support Set Accuracy (SSA)** SSA is the accuracy of the support set using an NCM classifier. We consider a few-shot task where the support set also plays the role of the query set.

**Support Set Confidence (SSC)** SSC is a confidence score. It is a soft version of SSA. It measures to what extent the shots are centered around their respective centroids.

$$SSC \simeq \mathbb{E}\left(\max_i\left(\mathrm{softmax}_k\left(\frac{-\mathbf{d}_{i,k}}{T}\right)\right)\right) \tag{2}$$

where $\mathbf{d}_{i,k}$ is the distance between a support sample $i$ and the different centroids $k$ and T the temperature. In a way it measures if the class shots are well grouped.

**Monte-Carlo Sampling (MCS).** In MCS, we measure an empirical covariance matrix and centroid for each class of the support set in the feature space. We then generate virtual data points in the feature space that mimic the distribution of the support set. These data points are classified the same cay any query would and the derived accuracy is used as a proxy to the actual one. In the case of a single shot an isotropic variance is used.

**Rank-Me (RKM).** RKM is another heuristic correlated with the performance of future downstream tasks introduced in (Garrido et al., 2022). The idea is to define a soft version of the rank (Roy & Vetterli, 2007) and measure this pseudo rank on a matrix of features from a model. The higher the rank the higher the performance. We use the the features of the support set to compute it.

**FIM** FIM corresponds to the Fischer Information Matrix as described in Achille et al. (2019). We have directly made use of their code to create embeddings for tasks and datasets. The Fisher information matrix (F) is defined as

$$F = \mathbb{E}_{x,y\sim\hat{p}(x)p_\theta(y|x)}\left[\nabla_\theta \log p_\theta(y|x)\nabla_\theta \log p_\theta(y|x)^T\right] \tag{3}$$

that is, the expected covariance of the scores (gradients of the log-likelihood) with respect to the model parameters $\theta$. $x$ are inputs and $y$ are labels. $\hat{p}$ is the empirical distribution defined by the training set and $p = \sigma \circ g \circ h_\theta$ the baseline model. The distance is measured using the normalized diagonals of the FIM of datasets. Cosine distance is used. We used the probe networks proposed by (Achille et al., 2019). We compute the distances between clusters of base classes and support sets from few-shot tasks.

**AA** Average Activation simply selects the cluster of classes the most activated by the support set of a task. The index of the selected cluster $s$ is :

$$s = \operatorname*{argmax}_i\left(\sum_{c\in\mathcal{P}_i; x\in\mathcal{S}} p_\theta(y_c|x)\right) \tag{4}$$

where $p_\theta(y_c|x)$ is the activation of the base class $c$ for a given image $x$ in the support set $\mathcal{S}$. $\mathcal{P}_i$ is the cluster of base classes $i$.

## A.5 DISCUSSION ON THE DIFFERENCE BETWEEN DATASETS

Here, we try to explain why we observe differences in performance boost across datasets. We mentioned different explanations in the paper.

We ruled out the idea that boosts are only obtained for *easy tasks* (high accuracy). Such tasks would only be found in datasets such as CUB and not on Traffic Signs or Aircraft where the domain gap with ImageNet is larger. To do this, we invoked the inverse correlation of accuracy with boost of performance. We provide empirical evidence of this in the Figure 17.

We briefly discussed the differences between Imagenet and the target datasets in the paper. Figure 10 helps us to understand the results of the first table in the paper.

**Aircraft**  Relatively poor boosts are obtained on the Aircraft dataset. Aircraft is almost fully captured by two base classes. The 'airliner' class could be the source of *collapse* of all Aircraft classes. Such collapse would create confusion between aircraft classes. This would mean that paradoxically closely related classes could be detrimental to the performance. Further research is required on this point. Almost all *Aircraft* images are typical airliner. Contrarily, we find that DTD and CUB cover a wide variety of closely related classes.

**Traffic Signs**  Finally, Traffic Signs contains only very low-resolution, tightly cropped images and is thus particularly different from ImageNet. It appears across all experimental settings that any fine-tuning seems detrimental to the few-shot performance on Traffic Signs.

## A.6 LOGISTIC REGRESSION FEW-SHOT CLASSIFIER

We verified that our method also provides better features when using a Logistic Regression (LR) in place of the NCM classifier. Table 2 clearly shows the consistent and positive impact of our DI finetuning on the representations even when using this new classifier.

Table 2: Accuracy obtained using a Logistic Regression (LR) classifier for both the baseline and proposed methodology (instead of NCM).

| Dataset | 1-shot 5-ways | | 5-shot 5-ways | | MD | |
|---|---|---|---|---|---|---|
| | Baseline | $\Delta$ | Baseline | $\Delta$ | Baseline | $\Delta$ |
| Aircraft | 42.28 ±0.72 | +0.07 ±0.42 | 68.50 ±0.69 | -0.08 ±0.33 | 72.47 ±0.99 | +0.88 ±0.27 |
| CUB | 65.28 ±0.82 | +4.81 ±0.47 | 86.74 ±0.58 | +2.91 ±0.29 | 77.85 ±0.93 | +2.81 ±0.21 |
| DTD | 50.57 ±0.78 | +1.88 ±0.50 | 72.02 ±0.60 | +2.15 ±0.37 | 80.21 ±0.74 | +1.92 ±0.31 |
| Fungi | 55.58 ±0.90 | +0.45 ±0.45 | 76.32 ±0.76 | +1.47 ±0.33 | 42.78 ±1.09 | +2.59 ±0.27 |
| Omniglot | 66.41 ±0.99 | +2.77 ±1.14 | 87.68 ±0.63 | +1.86 ±0.66 | 64.45 ±1.35 | +3.33 ±0.61 |
| MSCOCO | 46.97 ±0.87 | +0.83 ±0.41 | 66.57 ±0.73 | +0.60 ±0.32 | 44.42 ±1.09 | +1.38 ±0.18 |
| Traffic Signs | 62.36 ±0.85 | -1.16 ±0.93 | 83.96 ±0.64 | -1.18 ±0.61 | 59.73 ±1.12 | +2.78 ±0.38 |
| VGG Flower | 80.11 ±0.72 | +1.50 ±0.38 | 95.15 ±0.30 | +0.63 ±0.19 | 91.19 ±0.61 | +1.91 ±0.19 |
| **Average** | 58.69 ±0.44 | +1.39 ±0.23 | 79.62 ±0.35 | +1.04 ±0.15 | 66.64 ±0.58 | +2.20 ±0.12 |

## A.7 TRAINING FROM SCRATCH

We compared the DI finetuning to training from scratch on the same DI subsets. We report the results in Table 3. We see that only for omniglot, training on fewer, more similar classes, helps. We used the same hyperparameters to train our baseline model.

Table 3: Accuracy obtained when deploying the proposed methodology training from scratch on DI subsets (instead of finetuning).

| Dataset | 1-shot 5-ways | | 5-shot 5-ways | | MD | |
|---|---|---|---|---|---|---|
| | Baseline | $\Delta$ | Baseline | $\Delta$ | Baseline | $\Delta$ |
| Aircraft | 39.95 ±0.70 | -7.77 ±0.62 | 63.18 ±0.74 | -19.12 ±0.64 | 65.87 ±0.90 | -25.28 ±0.60 |
| CUB | 64.34 ±0.90 | -8.13 ±0.85 | 87.78 ±0.59 | -9.58 ±0.53 | 79.29 ±0.90 | -14.43 ±0.42 |
| DTD | 45.21 ±0.77 | -1.24 ±0.76 | 70.10 ±0.60 | -6.98 ±0.53 | 76.03 ±0.69 | -8.83 ±0.53 |
| Fungi | 53.01 ±0.92 | -11.25 ±0.78 | 74.87 ±0.79 | -15.54 ±0.61 | 51.57 ±1.16 | -15.87 ±0.50 |
| Omniglot | 61.80 ±1.03 | +3.10 ±1.26 | 81.53 ±0.76 | +2.85 ±0.84 | 59.51 ±1.31 | +3.82 ±0.66 |
| MSCOCO | 43.91 ±0.85 | -5.52 ±0.62 | 63.04 ±0.79 | -9.39 ±0.58 | 44.99 ±0.99 | -10.37 ±0.35 |
| Traffic Signs | 57.35 ±0.85 | -5.17 ±1.00 | 74.11 ±0.78 | -4.17 ±0.77 | 53.77 ±1.05 | -5.03 ±0.46 |
| VGG Flower | 75.86 ±0.84 | -8.80 ±0.82 | 94.46 ±0.33 | -6.94 ±0.46 | 92.77 ±0.58 | -8.63 ±0.40 |
| **Average** | 55.18 ±0.44 | -5.60 ±0.33 | 76.13 ±0.38 | -8.61 ±0.29 | 65.47 ±0.55 | -10.58 ±0.29 |

## A.8 SILHOUETTE SCORES

We provide in Table 4 silhouette scores (Rousseeuw, 1987) (from scikit-learn) which highlight how target classes are more separable thanks to the DI finetuning. It provides insight into how well each sample lies within its class, which is a reflection of the compactness and separation of the classes. Across all datasets, the silhouette score of features in the different classes increased by $\Delta = +0.0103$ with an average silhouette score for the baseline of -0.001.

| Dataset | Baseline | Ours |
|---|---|---|
| Aircraft | 0.0104 | 0.0096 |
| CUB | 0.0303 | 0.0317 |
| DTD | 0.0293 | 0.0460 |
| Fungi | -0.0436 | -0.0342 |
| Omniglot | -0.0719 | -0.0016 |
| Traffic Signs | -0.0279 | -0.0374 |
| VGG Flower | 0.0919 | 0.0913 |
| MSCOCO | -0.0277 | -0.0224 |
| **Average** | -0.0011 | 0.0103 |

Table 4: Comparative Analysis of Silhouette Scores for Features Extracted Using Two Different Backbones Across Diverse Datasets.

## A.9   ABLATION STUDY ON THE NUMBER OF SELECTED CLASSES

Figure 2: Relative gain in accuracy compared to the baseline after fine-tuning (Domain Informed setting), varying the number of classes $M$ selected using the Average Activation (AA) method. The star ticks correspond to the points where 90% of the cumulative activation across classes is reached. Apart from Aircraft and Fungi, the 90% cumulative activation threshold is reached around the same $M \sim 40$ which is around the peak of difference with baseline. Figure 10 shows the distribution of activation among classes.

## A.10   SEGMENTATION TASKS

Table 5 shows that our DI finetuning improved representations for segmentation tasks as well. This highlights the ability of our method to improve performances on different types of tasks.

Table 5: mIOU, mIOU reduced (hard classes ignored, specific to Cityscape) and accuracy on the segmentation dataset of Cityscape (Cordts et al., 2016) using the method developed in (Yang et al., 2021). Our experiments compare DI feature extractors with our baseline feature extractor on the same seeds (paired tests).

| Metric | 1-shot 5-ways | | 5-shot 5-ways | |
|---|---|---|---|---|
| | Baseline | DI | Baseline | DI |
| mIOU | $18.46 \pm 0.26$ | $18.72 \pm 0.25$ | $22.76 \pm 0.13$ | $23.07 \pm 0.13$ |
| mIOU Reduced | $21.87 \pm 0.31$ | $22.17 \pm 0.30$ | $26.92 \pm 0.15$ | $27.28 \pm 0.15$ |
| Accuracy | $70.49 \pm 0.38$ | $71.13 \pm 0.33$ | $74.24 \pm 0.14$ | $74.78 \pm 0.13$ |

## A.11   FEATURE SPACE DISTORTION OR BETTER FEATURES?

To investigate whether the improvement is due to mere distortion as opposed to a better representation, we conducted a set of experiments where the backbone was frozen and only an additional linear layer (with bias) was trained on the class subset. This linear layer can thus distort the feature space without fundamentally changing the representation.

The results are presented in Table 6. Overall, training just this layer decreases the accuracy of the NCM classifier, providing evidence that finetuning does yield an improved representation, rather than a simple distortion of the feature space.

Table 6: Difference in performance between using the features from the backbone directly vs. adding an extra linear layer, using DI subsets. We use the NCM classifier on top each time. The Mode column give more details about the extra layer that is placed just before the classification head. This extra layer is trained with the classification head in the same way as step 1. When "Finetune" is added to the mode we simply also apply step 2 (unfrozen backbone). "640x640" corresponds to a randomly initialized linear layer. "640x640 Res" corresponds to the same layer initialized at 0 with a skip connection (c.f. ResNet architecture).

| Dataset | Mode | 1-shot 5-ways | 5-shot 5-ways | MD |
|---------|------|---------------|---------------|-----|
| **Average** | 640x640 Finetuned Res | $+1.71 \pm 0.35$ | $-0.08 \pm 0.25$ | $-1.19 \pm 0.20$ |
| | 640x640 Res | $+0.59 \pm 0.35$ | $-2.18 \pm 0.24$ | $-3.66 \pm 0.20$ |
| | 640x50 | $-1.69 \pm 0.37$ | $-6.64 \pm 0.28$ | $-10.44 \pm 0.26$ |
| | Finetuned 640x50 | $-0.80 \pm 0.38$ | $-5.60 \pm 0.28$ | $-8.83 \pm 0.25$ |
| | 640x640 | $-0.32 \pm 0.37$ | $-3.77 \pm 0.26$ | $-5.82 \pm 0.23$ |
| Aircraft | 640x640 Finetuned Res | $-1.11 \pm 0.85$ | $-5.16 \pm 0.76$ | $-7.81 \pm 0.59$ |
| | 640x640 Res | $-1.10 \pm 0.84$ | $-7.10 \pm 0.73$ | $-10.85 \pm 0.61$ |
| | 640x50 | $-5.05 \pm 0.86$ | $-14.11 \pm 0.74$ | $-21.84 \pm 0.66$ |
| | Finetuned 640x50 | $-4.50 \pm 0.81$ | $-13.76 \pm 0.77$ | $-20.64 \pm 0.66$ |
| | 640x640 | $-2.40 \pm 0.87$ | $-10.51 \pm 0.73$ | $-15.48 \pm 0.60$ |
| CUB | 640x640 Finetuned Res | $+8.02 \pm 0.93$ | $+1.82 \pm 0.48$ | $+0.49 \pm 0.41$ |
| | 640x640 Res | $+9.37 \pm 0.93$ | $+1.24 \pm 0.48$ | $-0.41 \pm 0.46$ |
| | 640x50 | $+6.37 \pm 0.97$ | $-1.80 \pm 0.50$ | $-6.38 \pm 0.52$ |
| | Finetuned 640x50 | $+7.86 \pm 0.94$ | $-0.92 \pm 0.48$ | $-4.82 \pm 0.50$ |
| | 640x640 | $+9.13 \pm 0.96$ | $+0.51 \pm 0.49$ | $-1.73 \pm 0.47$ |
| DTD | 640x640 Finetuned Res | $+3.79 \pm 0.99$ | $+1.65 \pm 0.66$ | $-0.48 \pm 0.63$ |
| | 640x640 Res | $+2.02 \pm 0.97$ | $+0.62 \pm 0.66$ | $-2.05 \pm 0.61$ |
| | 640x50 | $+3.91 \pm 1.01$ | $-2.16 \pm 0.63$ | $-7.06 \pm 0.67$ |
| | Finetuned 640x50 | $+5.63 \pm 0.99$ | $-2.02 \pm 0.66$ | $-6.73 \pm 0.65$ |
| | 640x640 | $+2.74 \pm 1.02$ | $+0.11 \pm 0.65$ | $-3.07 \pm 0.62$ |
| Fungi | 640x640 Finetuned Res | $-0.54 \pm 0.99$ | $-3.09 \pm 0.67$ | $-2.82 \pm 0.56$ |
| | 640x640 Res | $-1.72 \pm 0.98$ | $-5.14 \pm 0.67$ | $-4.60 \pm 0.56$ |
| | 640x50 | $-2.53 \pm 0.98$ | $-8.89 \pm 0.69$ | $-10.24 \pm 0.60$ |
| | Finetuned 640x50 | $-2.60 \pm 0.98$ | $-8.25 \pm 0.67$ | $-8.03 \pm 0.58$ |
| | 640x640 | $-1.71 \pm 0.99$ | $-6.29 \pm 0.67$ | $-6.16 \pm 0.62$ |
| MSCOCO | 640x640 Finetuned Res | $+1.20 \pm 0.98$ | $+4.04 \pm 0.70$ | $+3.05 \pm 0.46$ |
| | 640x640 Res | $+1.81 \pm 0.99$ | $+1.81 \pm 0.75$ | $+0.59 \pm 0.45$ |
| | 640x50 | $+1.99 \pm 1.02$ | $+0.52 \pm 0.76$ | $-3.13 \pm 0.46$ |
| | Finetuned 640x50 | $+2.44 \pm 0.97$ | $+1.08 \pm 0.76$ | $-1.94 \pm 0.46$ |
| | 640x640 | $+1.80 \pm 0.98$ | $+1.17 \pm 0.77$ | $-0.81 \pm 0.45$ |
| Omniglot | 640x640 Finetuned Res | $-0.42 \pm 1.19$ | $-0.84 \pm 0.84$ | $-1.61 \pm 0.57$ |
| | 640x640 Res | $-5.16 \pm 0.88$ | $-6.10 \pm 0.55$ | $-7.43 \pm 0.49$ |
| | 640x50 | $-11.48 \pm 0.95$ | $-15.09 \pm 0.63$ | $-17.28 \pm 0.61$ |
| | Finetuned 640x50 | $-9.04 \pm 1.27$ | $-11.78 \pm 0.93$ | $-13.66 \pm 0.66$ |
| | 640x640 | $-7.93 \pm 0.95$ | $-9.24 \pm 0.58$ | $-10.78 \pm 0.50$ |
| Traffic Signs | 640x640 Finetuned Res | $+2.06 \pm 0.92$ | $+2.14 \pm 0.71$ | $+1.19 \pm 0.44$ |
| | 640x640 Res | $+1.02 \pm 0.95$ | $-0.00 \pm 0.70$ | $-0.95 \pm 0.43$ |
| | 640x50 | $-1.85 \pm 0.94$ | $-4.81 \pm 0.74$ | $-8.44 \pm 0.45$ |
| | Finetuned 640x50 | $-1.86 \pm 0.95$ | $-3.12 \pm 0.73$ | $-7.01 \pm 0.46$ |
| | 640x640 | $-0.06 \pm 0.94$ | $-0.87 \pm 0.72$ | $-2.32 \pm 0.43$ |
| VGG Flower | 640x640 Finetuned Res | $+0.65 \pm 0.91$ | $-1.17 \pm 0.38$ | $-1.54 \pm 0.33$ |
| | 640x640 Res | $-1.51 \pm 0.95$ | $-2.75 \pm 0.38$ | $-3.57 \pm 0.34$ |
| | 640x50 | $-4.83 \pm 0.96$ | $-6.81 \pm 0.47$ | $-9.19 \pm 0.43$ |
| | Finetuned 640x50 | $-4.35 \pm 0.93$ | $-5.99 \pm 0.47$ | $-7.81 \pm 0.41$ |
| | 640x640 | $-4.16 \pm 1.01$ | $-5.06 \pm 0.43$ | $-6.19 \pm 0.40$ |

## A.12 SUPPORT SET FINE-TUNING

Our fine-tuning on the support set followed as closely as possible the protocol described in (Triantafillou et al., 2019). They used a variety of configurations. We reproduced three of them (reported as best). Results are shown in 7. The performances are very low most of the time. Over-fitting is probably the issue on such small training dataset.

Table 7: Performance of fine-tuning on the support set with varying hyperparameters. We could not explore more than three settings as these require long computational effort. All Positive values are highlighted. Frozen signifies that only the last classification layer was trained while the rest of the network was frozen.

| Sampling | Dataset | Frozen; $lr = 10^{-3}$ | $lr = 10^{-3}$ | $lr = 10^{-4}$ |
|---|---|---|---|---|
| 1-shot 5-ways | Aircraft | -12.24 ±0.73 | -3.60 ±0.64 | -6.64 ±0.66 |
| | CUB | -18.04 ±0.86 | -19.28 ±0.88 | -25.52 ±0.97 |
| | DTD | -3.74 ±0.88 | **0.66 ±0.77** | -6.32 ±0.78 |
| | Fungi | -10.90 ±0.81 | -6.59 ±0.74 | -14.91 ±0.87 |
| | Omniglot | -29.13 ±1.06 | -3.16 ±1.11 | -21.17 ±1.08 |
| | MSCOCO | -4.47 ±0.70 | -5.44 ±0.66 | -9.09 ±0.70 |
| | Traffic Signs | -8.37 ±0.90 | -4.67 ±0.66 | -8.73 ±0.75 |
| | VGG Flower | -20.69 ±1.15 | **0.19 ±0.79** | -16.78 ±0.95 |
| | **Average** | -13.45 ±0.85 | -5.24 ±0.75 | -13.64 ±0.81 |
| 5-shot 5-ways | Aircraft | -24.55 ±0.85 | -1.48 ±0.61 | -11.71 ±0.67 |
| | CUB | -15.60 ±0.82 | -18.97 ±0.63 | -25.50 ±0.70 |
| | DTD | -16.33 ±0.84 | -3.12 ±0.59 | -9.49 ±0.62 |
| | Fungi | -13.86 ±0.81 | -8.33 ±0.62 | -18.28 ±0.76 |
| | Omniglot | -39.31 ±1.08 | **3.53 ±0.85** | -22.57 ±1.09 |
| | MSCOCO | -5.11 ±0.66 | -6.20 ±0.63 | -11.43 ±0.65 |
| | Traffic Signs | -4.03 ±0.81 | **6.17 ±0.62** | -0.67 ±0.62 |
| | VGG Flower | -17.36 ±0.96 | -1.45 ±0.37 | -9.87 ±0.57 |
| | **Average** | -17.02 ±0.81 | -3.73 ±0.59 | -13.69 ±0.68 |
| MD | Aircraft | -33.49 ±0.90 | **5.33 ±0.69** | -16.98 ±0.82 |
| | CUB | -18.49 ±0.65 | -14.51 ±0.60 | -39.36 ±0.80 |
| | DTD | -24.93 ±0.94 | -6.67 ±0.68 | -11.06 ±0.63 |
| | Fungi | -18.90 ±0.65 | -15.05 ±0.53 | -30.75 ±0.66 |
| | Omniglot | -40.25 ±1.02 | -4.59 ±1.07 | -36.27 ±1.01 |
| | MSCOCO | -8.85 ±0.44 | -17.00 ±0.72 | -20.21 ±0.50 |
| | Traffic Signs | -14.70 ±0.57 | **0.77 ±1.00** | -16.93 ±0.65 |
| | VGG flower | -34.71 ±1.05 | -5.18 ±0.51 | -25.93 ±1.02 |
| | **Average** | -24.29 ±0.76 | -7.11 ±0.71 | -24.69 ±0.74 |

## A.13 OTHER TABLES AND FIGURES

Table 8: Performance change using the fine-tuning on the support (S), with a Task-Informed (TI) subset selection, a Domain-Informed (DI) subset selection, and DI-UOT subset selection. All positive boosts with overlapping confidence intervals are bolded.

| Dataset | Method | 1-shot 5-ways | | 5-shots 5-ways | | MD | |
|---|---|---|---|---|---|---|---|
| | | Baseline | Δ | Baseline | Δ | Baseline | Δ |
| Aircraft | S | | -3.60 ±0.64 | | -1.48 ±0.61 | | **+5.33 ±0.69** |
| | TI | | -0.06 ±0.33 | | **+0.26 ±0.31** | | +1.33 ±0.25 |
| | DI | 39.95 ±0.70 | **+0.34 ±0.32** | 63.18 ±0.74 | **+0.54 ±0.31** | 65.86 ±0.90 | +1.32 ±0.27 |
| | DI-UOT | | -0.25 ±0.33 | | -0.04 ±0.30 | | +0.86 ±0.27 |
| | TI-UOT | | -0.06 ±0.33 | | -0.01 ±0.30 | | +0.93 ±0.26 |
| CUB | S | | -19.28 ±0.88 | | -18.97 ±0.63 | | -14.51 ±0.60 |
| | TI | | +2.64 ±0.44 | | **+2.16 ±0.26** | | +1.08 ±0.19 |
| | DI | 64.34 ±0.90 | **+3.27 ±0.44** | 87.78 ±0.59 | **+2.29 ±0.26** | 79.29 ±0.90 | **+2.20 ±0.20** |
| | DI-UOT | | +2.99 ±0.43 | | **+2.07 ±0.27** | | **+1.97 ±0.20** |
| | TI-UOT | | +2.64 ±0.44 | | +1.11 ±0.26 | | +0.96 ±0.19 |
| DTD | S | | +0.66 ±0.77 | | -3.12 ±0.59 | | -6.67 ±0.69 |
| | TI | | **+2.85 ±0.46** | | **+2.77 ±0.33** | | **+2.44 ±0.29** |
| | DI | 45.21 ±0.77 | **+2.90 ±0.48** | 70.10 ±0.59 | **+2.96 ±0.33** | 76.03 ±0.69 | **+2.78 ±0.31** |
| | DI-UOT | | **+2.26 ±0.51** | | **+2.62 ±0.34** | | **+2.82 ±0.32** |
| | TI-UOT | | **+2.85 ±0.46** | | **+2.44 ±0.32** | | **+2.82 ±0.32** |
| Fungi | S | | -6.59 ±0.74 | | -8.33 ±0.62 | | -15.05 ±0.53 |
| | TI | | **+0.92 ±0.39** | | **+1.67 ±0.30** | | **+1.07 ±0.26** |
| | DI | 53.01 ±0.92 | **+1.07 ±0.41** | 74.87 ±0.80 | **+1.89 ±0.29** | 51.57 ±1.16 | **+1.38 ±0.25** |
| | DI-UOT | | +0.74 ±0.40 | | **+1.46 ±0.29** | | **+0.91 ±0.25** |
| | TI-UOT | | **+0.92 ±0.39** | | **+1.51 ±0.28** | | +0.80 ±0.25 |
| Omniglot | S | | -3.16 ±1.11 | | **+3.53 ±0.85** | | -4.59 ±1.07 |
| | TI | | **+2.65 ±0.38** | | **+2.94 ±0.29** | | **+3.74 ±0.23** |
| | DI | 61.80 ±1.03 | **+3.52 ±1.22** | 81.53 ±0.76 | **+3.57 ±0.81** | 59.51 ±1.31 | **+3.93 ±0.61** |
| | DI-UOT | | -3.70 ±1.00 | | -5.02 ±0.68 | | -5.76 ±0.66 |
| | TI-UOT | | **+2.65 ±0.38** | | +2.58 ±0.82 | | +2.46 ±0.62 |
| MSCOCO | S | | -5.44 ±0.66 | | -6.20 ±0.63 | | -17.00 ±0.72 |
| | TI | | **+1.27 ±0.35** | | **+1.87 ±0.29** | | +1.85 ±0.17 |
| | DI | 43.91 ±0.85 | **+1.62 ±0.34** | 63.04 ±0.79 | **+2.09 ±0.30** | 44.99 ±0.99 | **+2.25 ±0.17** |
| | DI-UOT | | **+1.27 ±0.35** | | **+1.75 ±0.29** | | **+2.09 ±0.18** |
| | TI-UOT | | **+1.27 ±0.35** | | +1.30 ±0.28 | | **+2.05 ±0.18** |
| Traffic Signs | S | | -4.67 ±0.66 | | **+6.17 ±0.62** | | **+0.77 ±1.00** |
| | TI | | **-0.84 ±0.32** | | -1.22 ±0.25 | | -2.02 ±0.17 |
| | DI | **57.35 ±0.85** | **-0.79 ±0.95** | 74.11 ±0.78 | -1.48 ±0.77 | **53.77 ±1.05** | -1.82 ±0.44 |
| | DI-UOT | | **-0.48 ±0.33** | | -0.64 ±0.27 | | -1.26 ±0.18 |
| | TI-UOT | | **-0.84 ±0.32** | | -0.99 ±0.77 | | -1.33 ±0.43 |
| VGG Flower | S | | +0.19 ±0.79 | | -1.45 ±0.37 | | -5.18 ±0.51 |
| | TI | | **+2.04 ±0.40** | | **+0.64 ±0.18** | | **+1.03 ±0.16** |
| | DI | 75.86 ±0.84 | **+1.88 ±0.41** | 94.46 ±0.33 | **+0.52 ±0.18** | 92.77 ±0.58 | **+0.84 ±0.16** |
| | DI-UOT | | **+2.18 ±0.40** | | **+0.67 ±0.18** | | **+0.90 ±0.16** |
| | TI-UOT | | **+2.04 ±0.40** | | **+0.90 ±0.17** | | **+0.95 ±0.16** |
| **Average** | S | | -5.24 ±0.78 | | -3.73 ±0.61 | | -7.11 ±0.73 |
| | TI | | **+1.43 ±0.38** | | **+1.39 ±0.28** | | **+1.31 ±0.21** |
| | DI | | **+1.73 ±0.57** | | **+1.55 ±0.41** | | **+1.61 ±0.30** |
| | DI-UOT | | +0.63 ±0.47 | | +0.36 ±0.33 | | +0.32 ±0.28 |
| | TI-UOT | | **+1.43 ±0.36** | | **+1.10 ±0.44** | | **+1.21 ±0.32** |

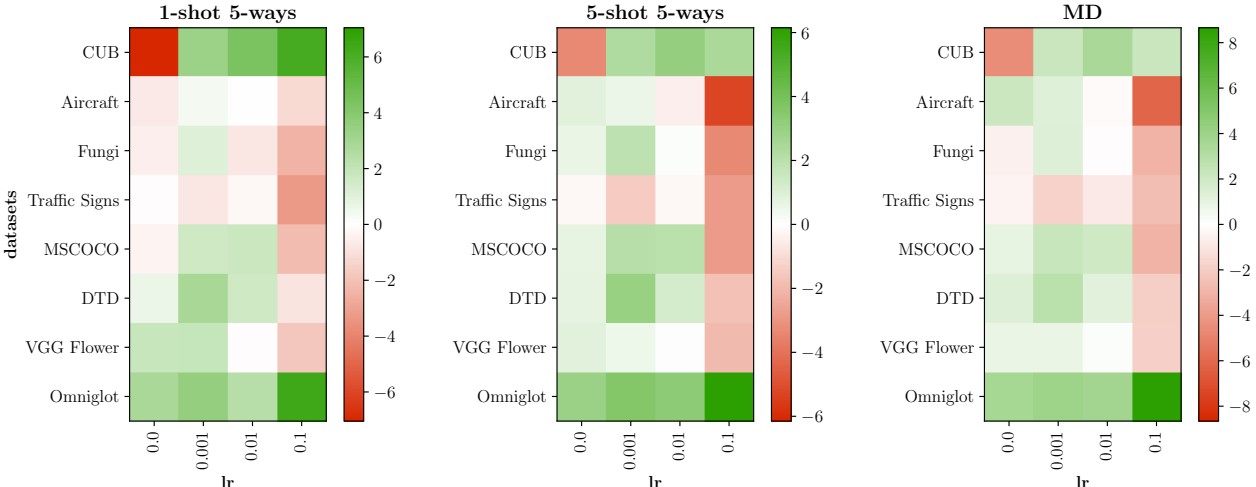

Figure 3: Boost in Accuracy compared to the baseline for various learning rates **lr** using the DI selected feature extractor of each dataset. Learning rate is set to 0 when only batch normalization statistics are updated. In the paper we only show the case of $lr = 0.001$. We observe a significant effect of the choice of the learning rate.

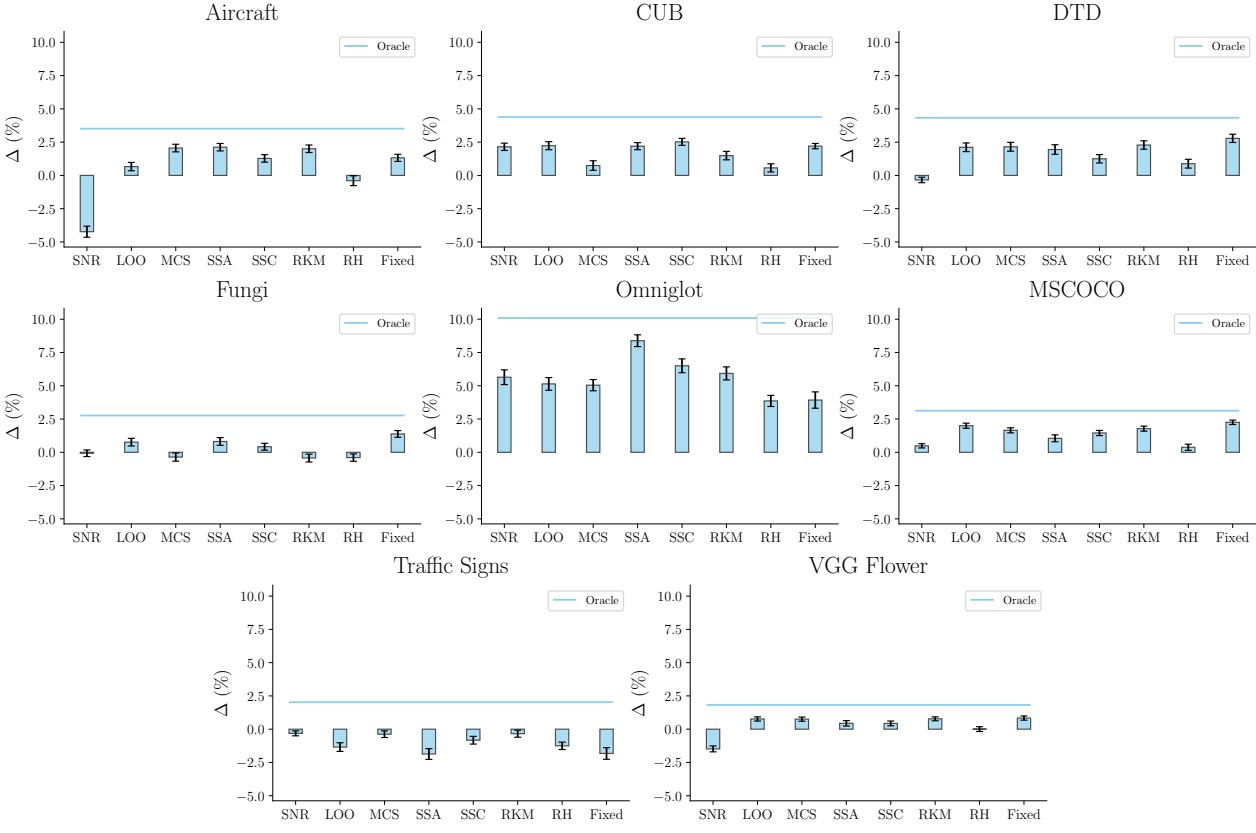

Figure 4: Selection of learning rate in DI setting using heuristics in MD sampling. Fixed corresponds to the performance of lr = 0.001 that was presented in the first table of the paper. Our methods outperforms the DI accuracy boost (Fixed) on Aircraft, Omniglot and Traffic Signs. We use the different learning rates presented in Table 3

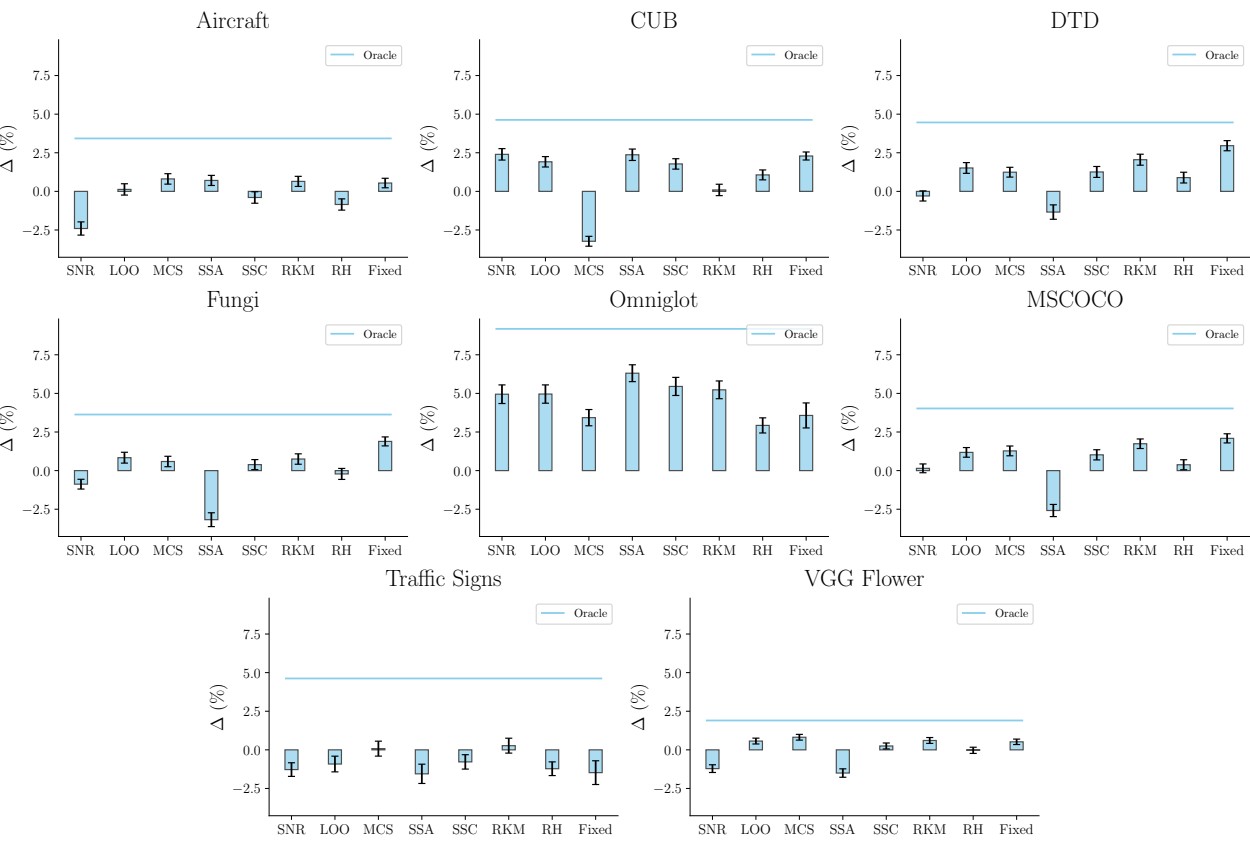

Figure 5: Selection of learning rate in DI setting using heuristics in 5-ways 5-shots sampling. Fixed corresponds to the performance of lr = 0.001 that was presented in the first table of the paper. Our methods outperforms the DI accuracy boost (Fixed) on Aircraft, Omniglot and Traffic Signs. We use the different learning rates presented in Table 3

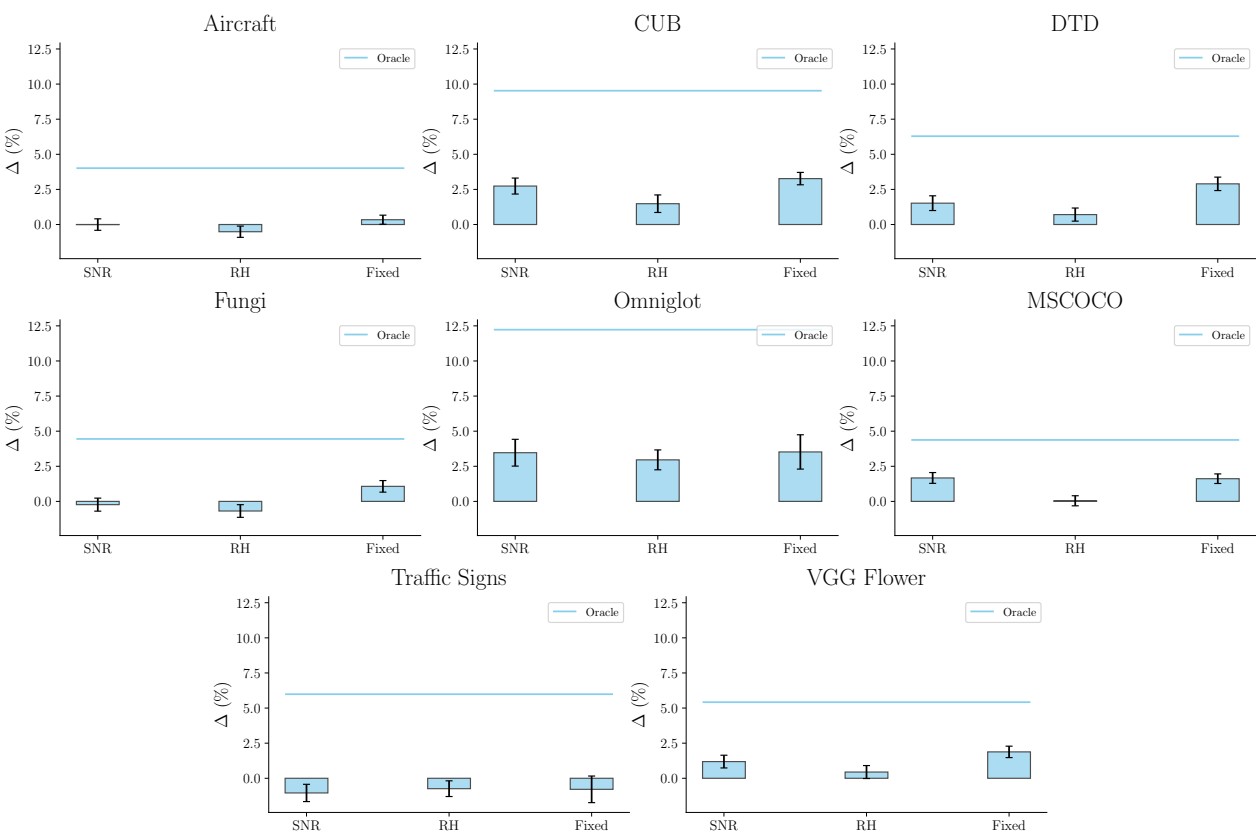

Figure 6: Selection of learning rate in DI setting using heuristics in 1-ways 5-shots sampling. Fixed corresponds to the performance of lr = 0.001 that was presented in the first table of the paper. In this case, the available data for the selection is not sufficient to outperform *Fixed*. We use the different learning rates presented in Table 3

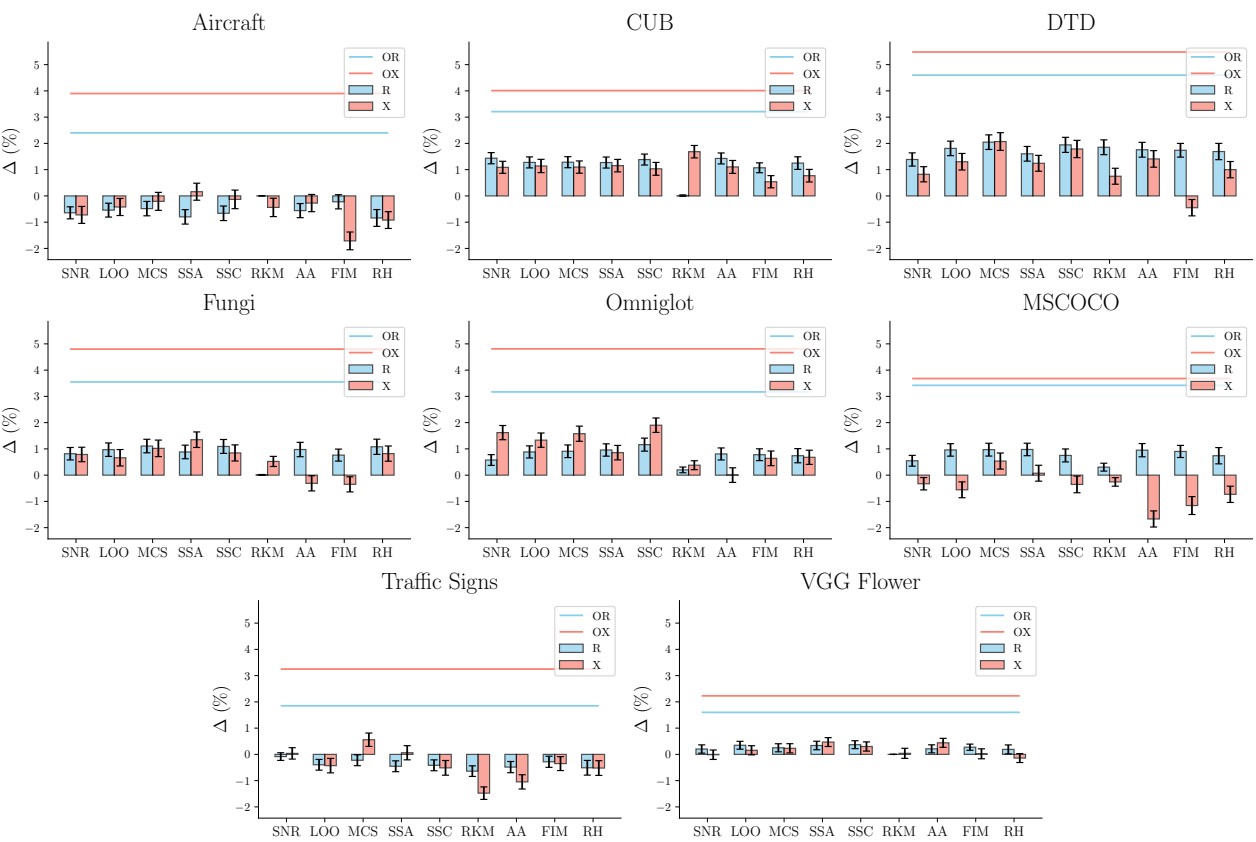

Figure 7: Difference of accuracy with baseline after feature extractor selection using heuristics. Task are sampled following the 5-ways 5-shots sampling procedure. In R (resp. X) heuristics select a feature extractor amongst the R (resp. X) library of feature extractor. The oracle OR (resp. OX) selects the best feature extractor for each task in the R (resp. X) library. The Random Heuristic (RH) picks a feature extractor uniformly at random.

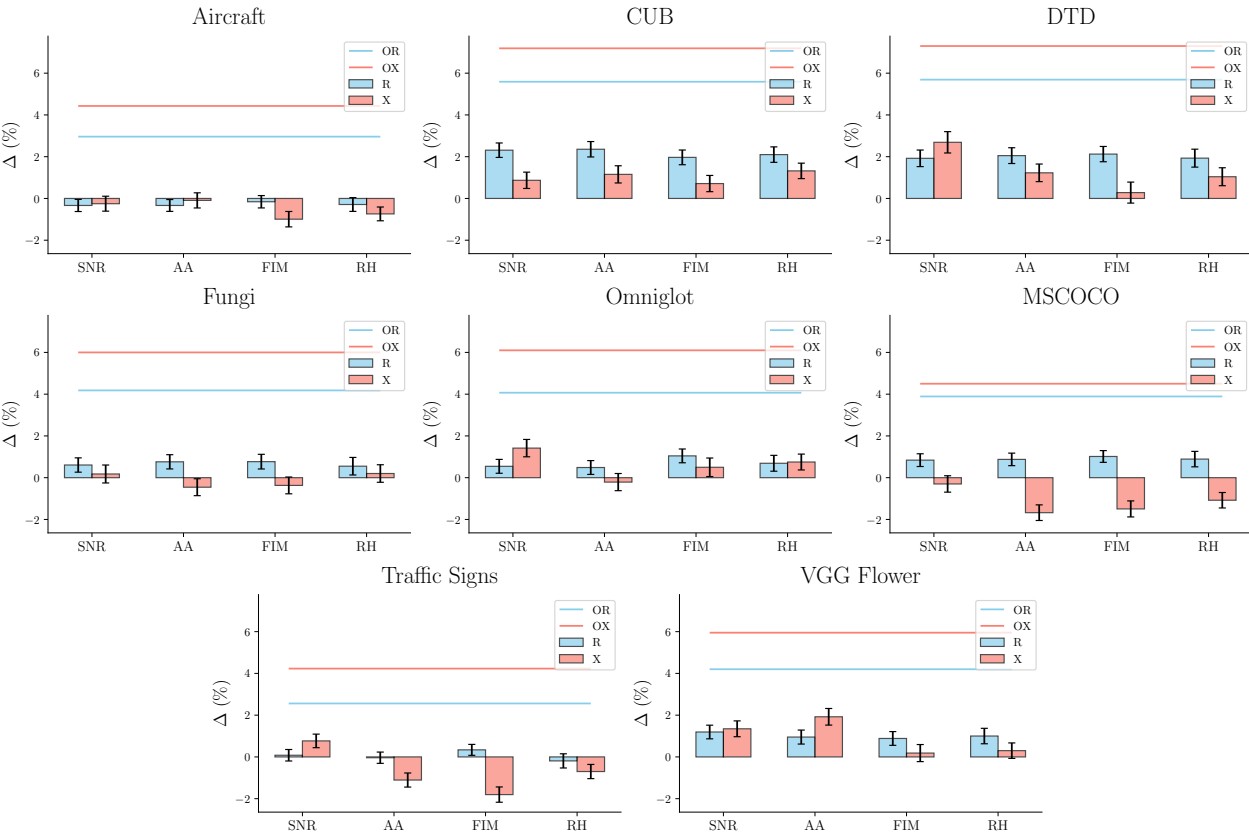

Figure 8: Difference of accuracy with baseline after feature extractor selection using heuristics. Task are sampled following the 1-ways 5-shots sampling procedure. In R (resp. X) heuristics select a feature extractor amongst the R (resp. X) library of feature extractor. The oracle OR (resp. OX) selects the best feature extractor for each task in the R (resp. X) library. The Random Heuristic (RH) picks a feature extractor uniformly at random.

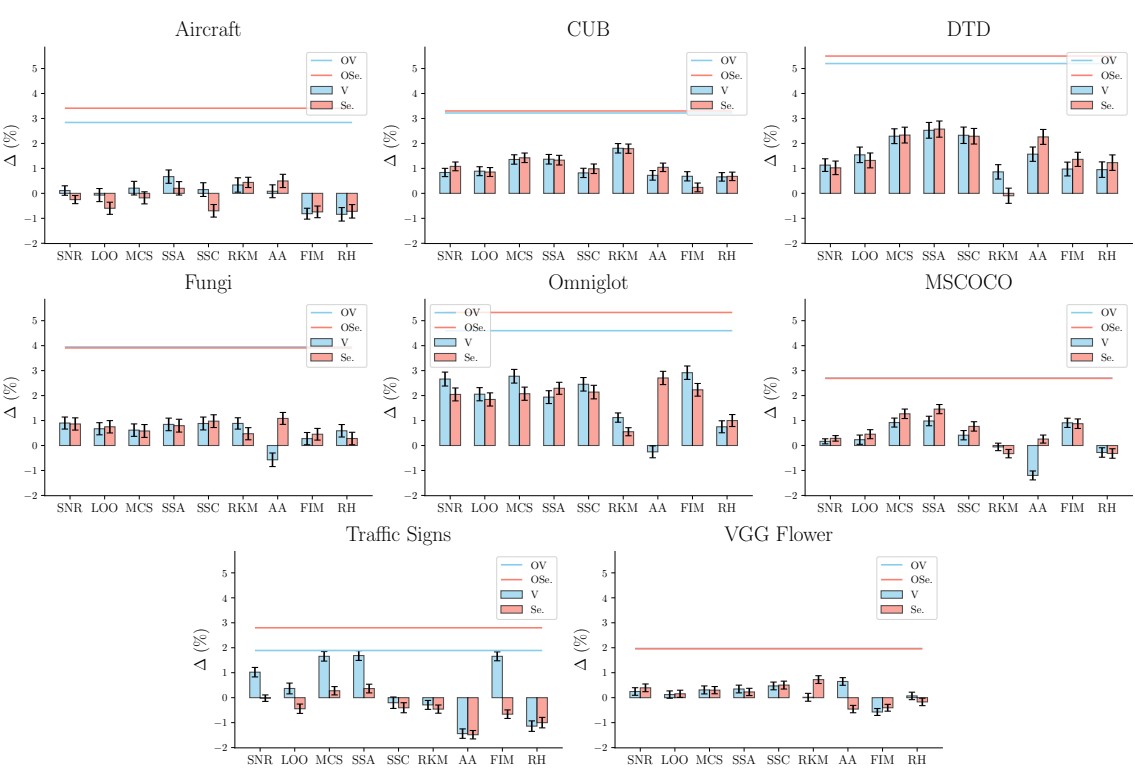

Figure 9: Difference of accuracy with baseline after feature extractor selection using heuristics. Task are sampled following the MD protocol. In V (resp. Se.) heuristics select a feature extractor amongst the V (resp. Se.) library of feature extractor. The oracle OV (resp. OSe.) selects the best feature extractor for each task in the V (resp .Se.) library. The Random Heuristic (RH) picks a random feature extractor.

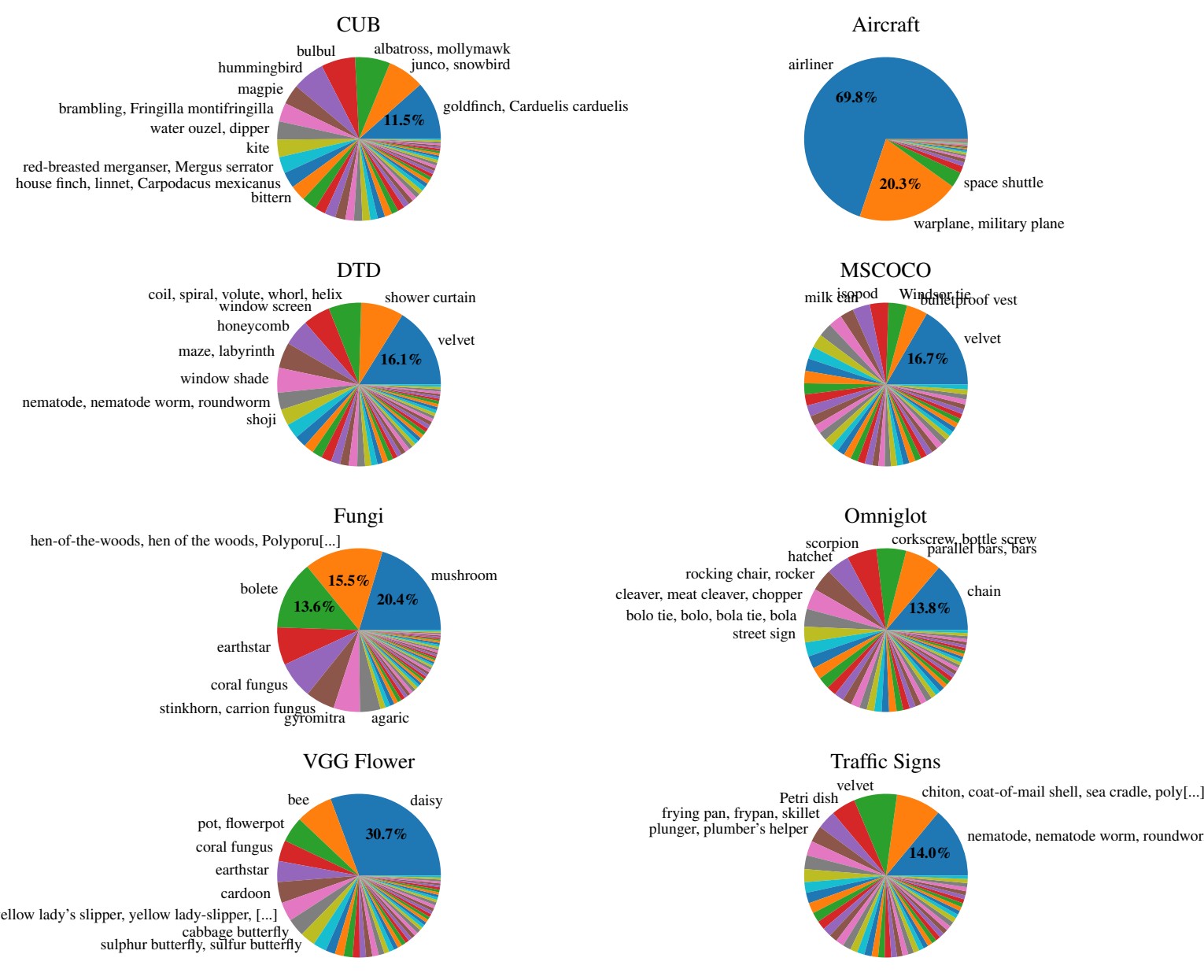

Figure 10: Logit activations of ImageNet classes when target datasets are processed by the base model. While it may seem surprising that the ImageNet 'Street sign' class is not strongly activated within the Traffic Signs dataset, this is because its tightly cropped, low resolution images are highly dissimilar from the photographs of street signs in ImageNet. Notice how Aircraft is almost fully captured by two classes.

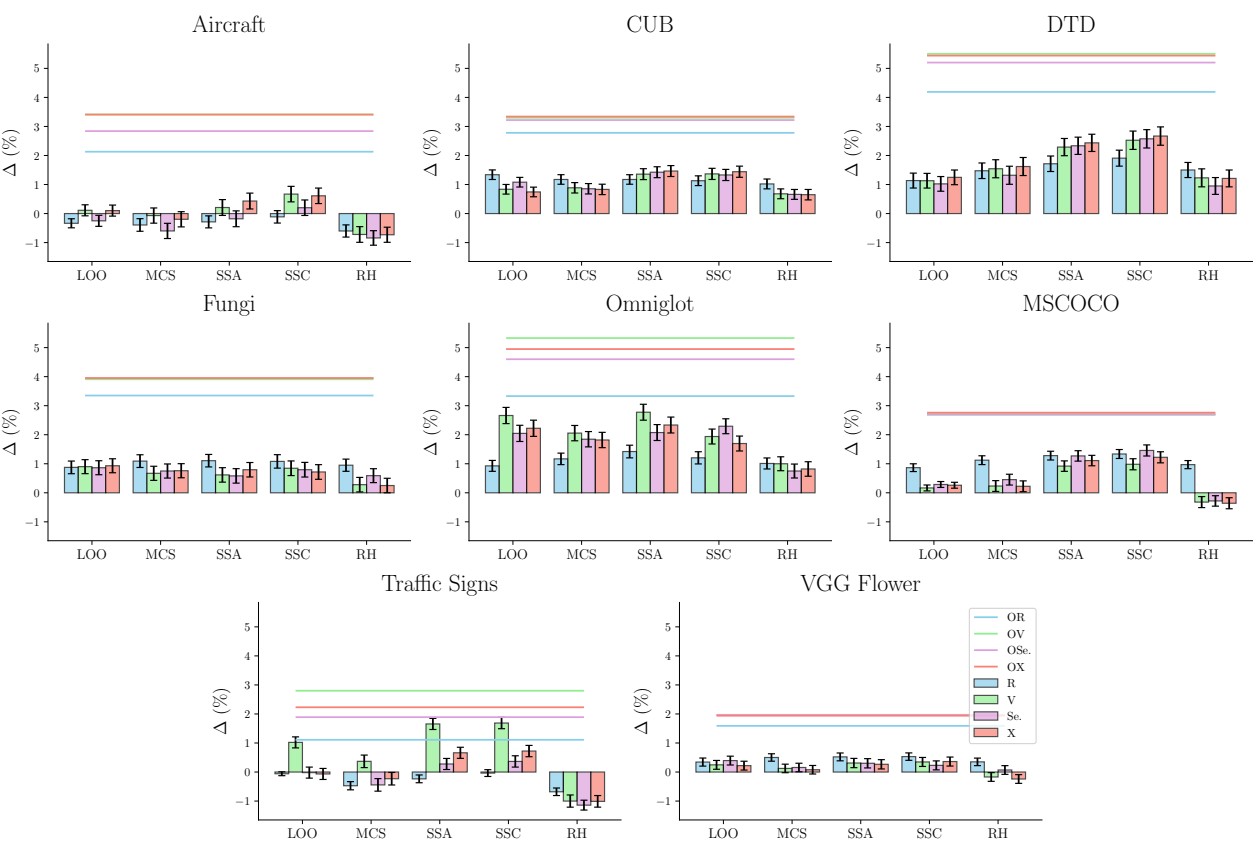

Figure 11: Ablation of the effect of R, V, Se. and X on a reduced number of heuristics for lisibility. X is sometimes outperformed by V and Se. but overall X is the best.

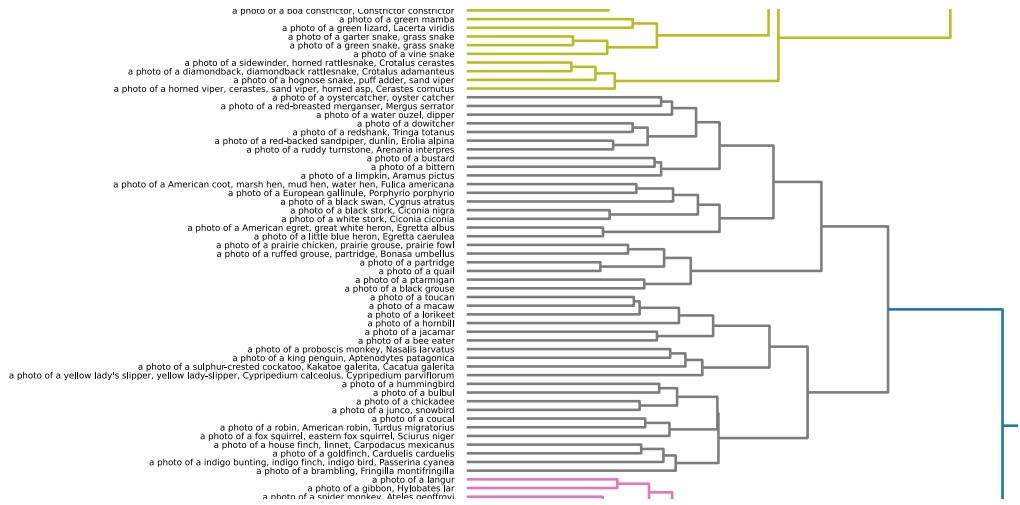

Figure 12: Zoom over the birds (gray), reptiles (yellow) and monkeys (pink) of the Semantic (Se.) dendrogram of classes built using Ward's method. Notice that we used "a photo of a" in front of each classes to improve the CLIP embedding. Out of the 44 classes in the birds cluster 3 classes are not birds : The proboscis monkey, Yellow Lady's Slipper, Fox Squirrel. Their semantic relations to birds must explain their relation to this cluster. Some ambiguous words like "Crane" or "Kite" might not be captured by the semantic embeddings.

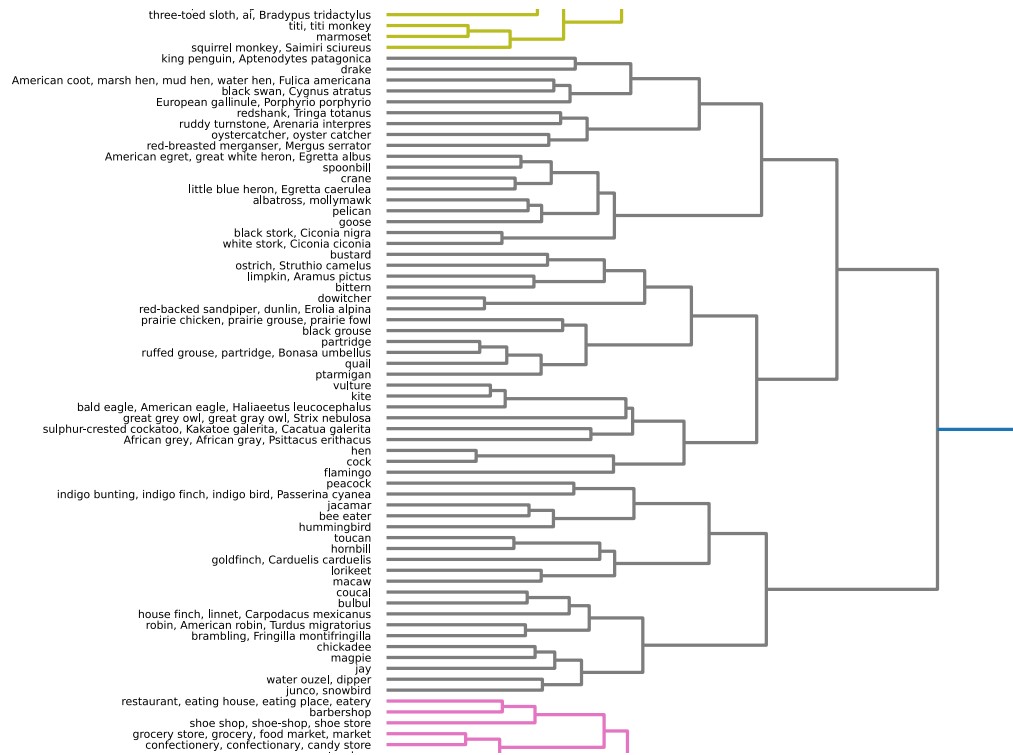

Figure 13: Zoom over the birds (gray), reptiles (yellow) and buildings (pink) of the Visual (Se.) dendrogram of classes built using Ward's method. Notice that "Kite" was classified as part of the birds for trivial reasons (kites in the sky can be mistaken for a bird).

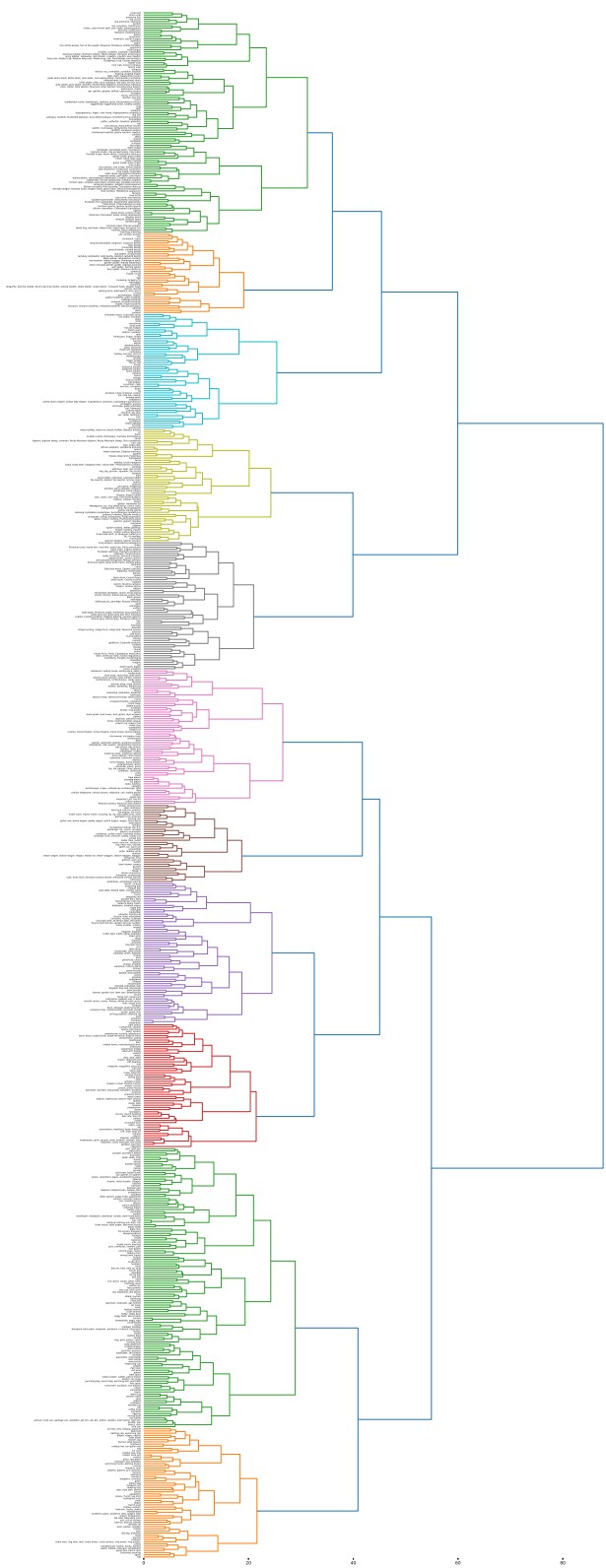

Figure 14: Visual (V). dendrogram of classes built using Ward's method

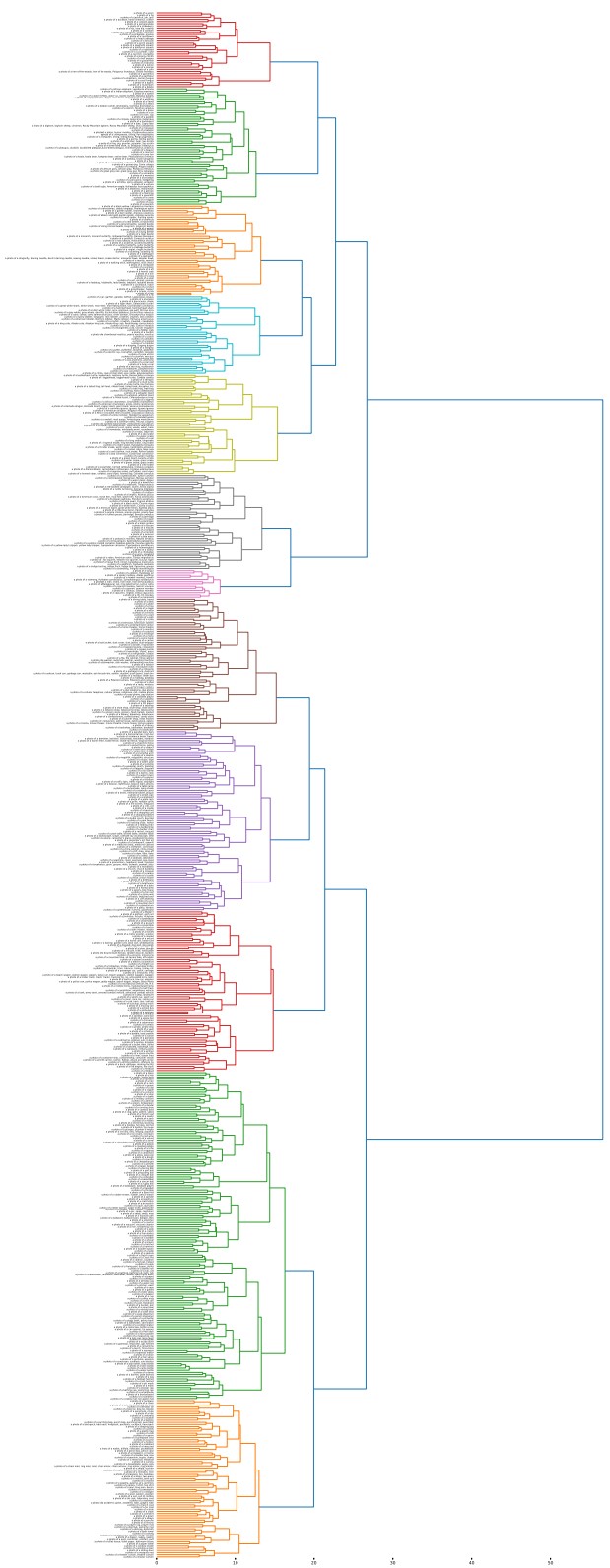

Figure 15: Semantic (Se.) dendrogram of classes built using Ward's method

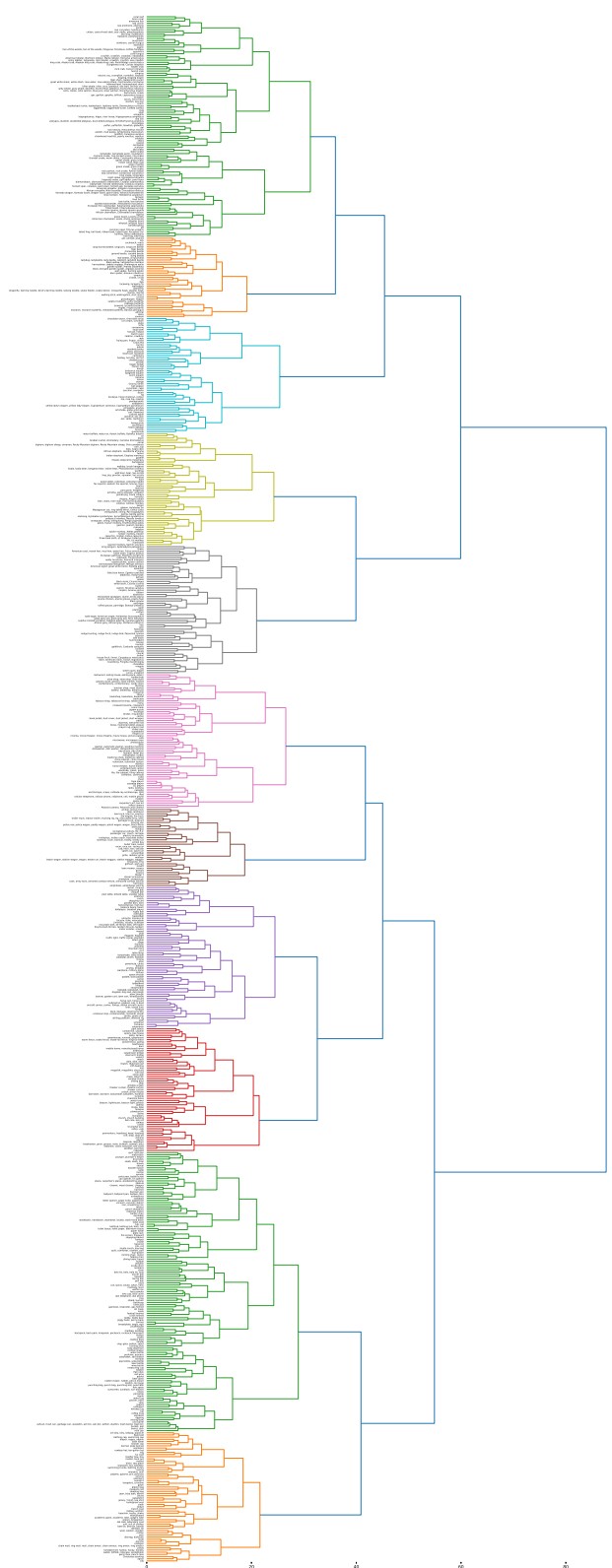

Figure 16: Visual-Semantic (X). dendrogram of classes built using Ward's method

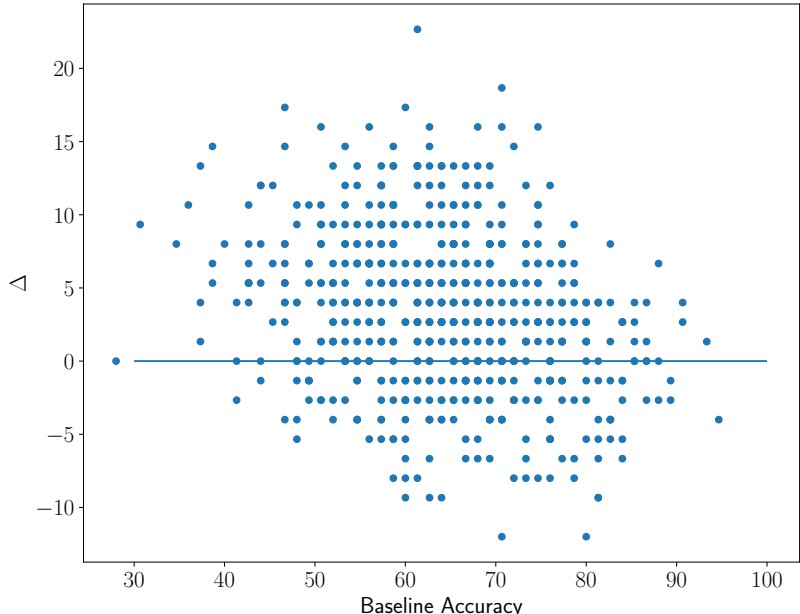

Figure 17: DI Boost versus baseline accuracy in 1-shot 5-ways for CUB. This refutes the hypothesis that only problems which already enjoy high accuracy can benefit from subset selection: rather, there is a negative correlation between boost in accuracy and baseline accuracy (as mentioned in the paper). The regular grid stems from the discrete set of possible outcomes for 75 query examples (5 ways with 15 query examples per class).

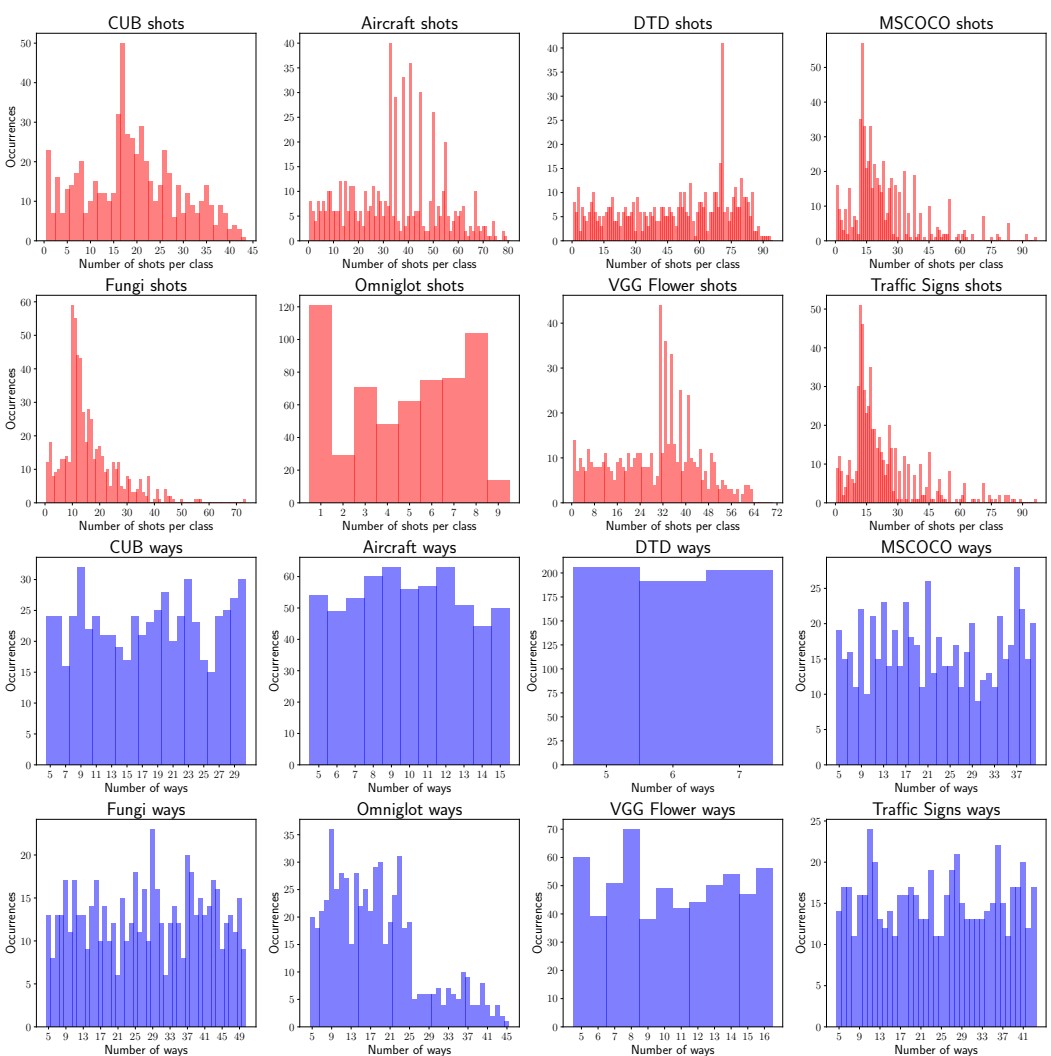

Figure 18: Histogram of the number of shots and ways for each dataset using MD sampling. This shows the great variability of the sampling procedure described in (Triantafillou et al., 2019).

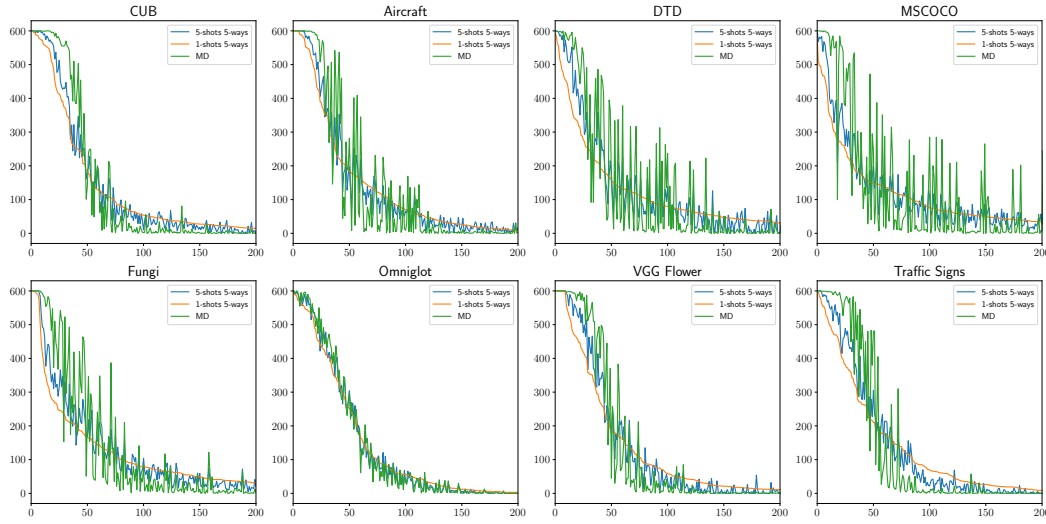

Figure 19: Number of selections of ImageNet1k classes. The classes are ordered to be less and less selected in 1-shots 5-ways. We observe a strong difference in class selection between samplings. 1-shot 5-ways is clearly less consistent across episodes since the initial plateau depicting the base classes which are selected in all 600 episodes (top left of each plot) is often much smaller (if present at all) for these tasks.

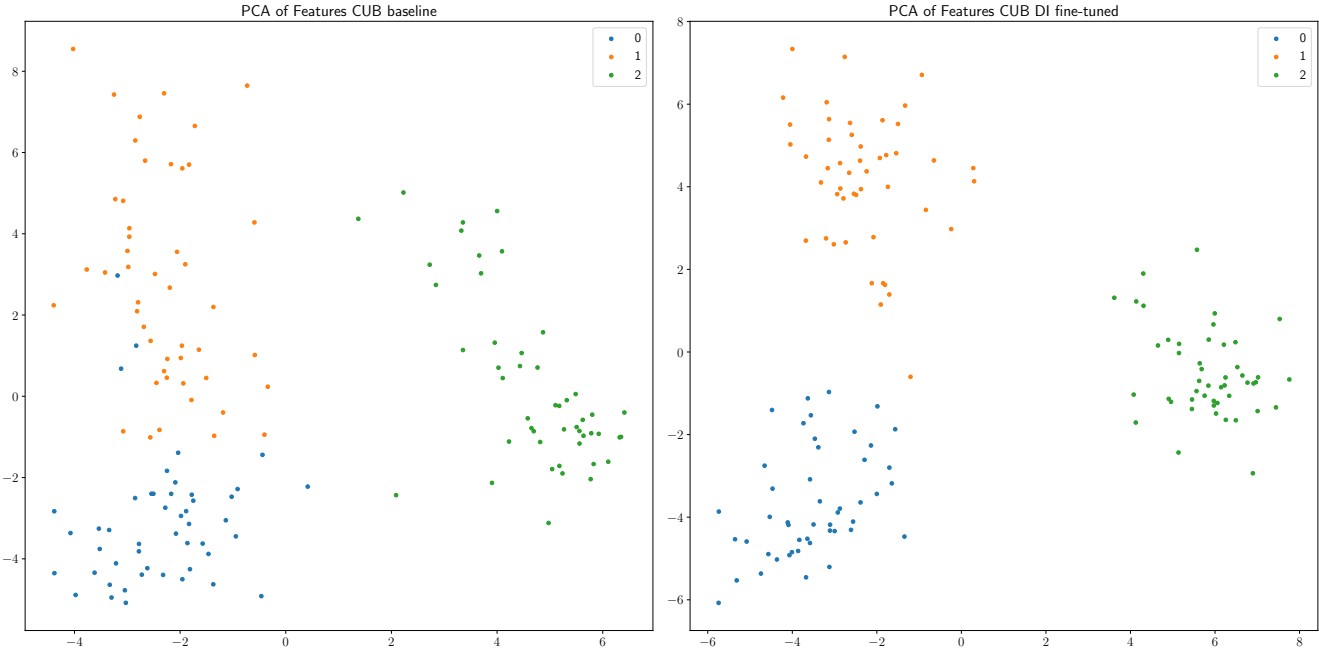

Figure 20: PCA of features of three classes before and after fine-tuning using DI (successful example). We clearly observe an increased separability between classes orange and blue.

