# OpenReview forum: "Few and Fewer: Learning Better from Few Examples Using Fewer Base Classes"
_ICLR.cc/2024/Conference — Submitted to ICLR 2024_

### Official Review · Reviewer_snfD · 2023-11-01

**Soundness:** 3 good
**Presentation:** 4 excellent
**Contribution:** 3 good
**Rating:** 6
**Confidence:** 4

**Summary:**

The authors raise and investigate the idea that fine-tuning a pretrained model on a subset of the (base) classes it has originally been trained on might yield beneficial results by potentially reducing the domain gap between train and target distributions in few-shot settings. The paper presents approaches for three different ‘levels’ of available target information, validated by experiments on several datasets and a variety of different heuristics for the case of least available information (aka uninformed setting). The authors demonstrate that a careful selection can indeed improve results across most benchmarks, but equally point out and discuss potential challenges when domain gaps are relatively big.

**Strengths:**

### Originality & Significance:
-  While the underlying idea of using a ‘better fitting’ subset of classes seems quite straight-forward and intuitively makes sense to reduce potential domain gaps, I am not aware of any such detailed study of this aspect;
&rarr; To me, the authors provide insights that are largely orthogonal to most recent developments in the few-shot area, honestly discuss their insights and provide open questions that could be tackled by the community in the future – therefore satisfying both ‘originality’ and ‘significance’.

### Quality:
- Realization and thorough investigation of a straight-forward and simple but very neat idea in a solid manner, using three different levels of available information (w.r.t target task/domain)
- Experiments conducted with a good selection of comparative methods to gauge performance improvements: representative baselines in the informed settings, as well as different heuristics in the uninformed setting -- including random selection and oracle upper bound
- The authors honestly discuss both their ‘achievements’ as well as potential challenges that exist with their approach, offering insights and intuitions what could potentially be tackled in future work;
### Clarity:
- The paper is very well written and easy to read and follow; The topic is well motivated with clear presentation throughout
- The authors do a great job of clearly stating the objectives at several stages throughout the paper, combined with the underlying intuitions as well as on-point discussion of the relevant background and results;

**Weaknesses:**

**TLDR;** I do not see any severe ‘prohibitive’ weaknesses in this work but have some concerns listed in the following; If the authors can address my questions & concerns, I’m happy to further increase my score and support (full) acceptance.

**Pre-training motivation**:
- The authors state that pre-training on the base classes “may even have a deleterious effect if the domain gap is too large” – and hence investigate if *one can reduce the domain gap by fine-tuning on a subset*:
&rarr;  While I do understand that this is ‘merely’ used as motivation, I somewhat doubt that in cases where the domain gap is so large that it (significantly) harms transfer learning, fine-tuning on a subset of these classes would help? (i.e. the presented approach) – some comments/insights regarding this would be helpful;

- Along the same lines: Given this motivation, wouldn’t then a reasonable baseline to compare to be the network ‘just’ trained on the subset of classes from scratch? In other words, how important is the pre-training on all the base classes? (as stated, it might even be harmful?)

**Slightly limited scope**:
- The presented results are somewhat constrained to the authors’ own setup – i.e. one baseline architecture that is used and then improved; However, the authors do not provide any indication as to what ‘sota’ methods that use the same backbone currently achieve on the chosen datasets;
&rarr;  *Note*: I do in no way expect the authors to ‘beat’ any sota method as this paper is mainly about providing insights, but it would be interesting for the reader to know how far the ‘plain’ baseline and the ‘improved’ one are from top-performing methods; Some insight whether this sub-class finetuning setup might help in other methods as well (e.g. during meta-finetuning) would also be helpful to further support the paper’s findings.

**Missing ablation**:
- The authors note that they choose the top 50 classes (AA) in the informed settings to create the subsets, mainly to keep ‘similar size of subsets’ to the uninformed settings;
&rarr; How important is this selection to achieve ‘good’ results? (robustness) – Some ablation would be helpful here;
&rarr; It feels like this might be highly dependent on the composition of the dataset (base classes)?
&rarr; I suspect there might be a more ‘generalizable’, dynamic and potentially better justifiable way to select the classes than a mere “top x” constant, e.g. certain % of the total mass / classes that cumulatively achieve certain threshold in the softmax; or even just a threshold on the ‘minimal contribution within the softmax’.

**Questions:**

Please see the "weaknesses" section for ‘main’ concerns;
The following are mostly questions to gain some further intuition & suggestions for improving the manuscript:

- Judging by the result presented in Fig 1, the random subsets seem to be surprisingly better when using AA, FIM and RH – do the authors have any insight/suspicion as to why this might be the case?
- Fig 8 (Appendix) indicates that the Semantic features generally provide better ‘upper bounds’ as shown by the oracle; Do the authors have some intuition whether that’s due to the ‘more powerful’ underlying CLIP training, or whether semantics (text) generally provide richer features for such selection? (if one has to be chosen)

**Comments/Suggestions**:
- Sec 4.2: The authors state that “MCS […] consistently led to a positive impact”, leading to the “ability to deploy such solutions in applications where strong constraints apply in terms of computation and/or latency” – However, doesn’t MCS actually have quite high compute requirements? (A fact that is also stated in the limitations and therefore somewhat contradictory)
- Sec 4.1: Explanation and/or reference for the meaning/significance of the silhouette score would be helpful to the reader (can be in appendix)
- Sec 4.2: “SNR […] yields *optimal* boost” -> potentially change wording, as “optimial” to me would mean it achieves the oracle performance

**Typos**:
- Sec 2 Terminology: “[…] techniques could be extended to the transductive setting is possible” -> Remove “is possible”
- Sec 2 Leightweight: second last line misses a ‘period’ after the meta-learning references .. Requeima et al., 2019) “.”
- Sec 4.2: “Figure 19” -> likely “Figure 1”?
- Sec 4.2 last line: “This experiments” -> plural/singular mixup

---

> ### Author Response · Authors · 2023-11-21
>
> We would like to thank the reviewer for their positive and constructive comments. In the following, we address their concerns and questions to the best of our ability.
>
> > I somewhat doubt that in cases where the domain gap is so large that it (significantly) harms transfer learning, fine-tuning on a subset of these classes would help?
>
> We would like to thank the reviewer for raising this point. Indeed, there exist obvious conditions where the domain gap can be reduced by selecting a subset of base classes, for example if the target task is precisely this subset of classes.
> In conditions where the target distribution is far from the base distribution, such as Omniglot compared to ImageNet, the intuition behind the approach is not as evident.
> Nevertheless, our experiments show that, even in the case of this large domain gap, the proposed methodology can significantly improve performance over the baseline, providing at least empirical evidence to support the idea that even in those cases fine-tuning on a subset of classes would help. We enriched the Methodology part of the main paper to reflect these points.
>
> > [...] wouldn’t then a reasonable baseline to compare to be the network ‘just’ trained on the subset of classes from scratch?
>
> We would like to thank the reviewer for suggesting this experiment, which indeed makes a lot of sense. We chose fine-tuning rather than training from scratch due to the computational overhead, especially in the Task-Informed setting. We ran a number of experiments to compare with training from scratch in the Domain-Informed setting (Table 3). While Omniglot did show an improvement when trained from scratch, most domains achieved worse accuracy compared to fine-tuning. However, this may simply be because the hyper-parameters were optimized for the large base dataset. In our view, this does not detract from the finding that fine-tuning on a subset of the base dataset improves the representation for a given task. We added these results to the appendix.
>
> > The presented results are somewhat constrained to the authors’ own setup [...]
>
> We have run new experiments using logistic regression instead of the NCM classifier on top of our existing backbones. This approach is supposedly close to SOTA according to a recent paper A Closer Look at Few-shot Classification Again by Luo et al. Our results (Table 2) show a very similar trend to those we obtained using the NCM classifier. As such, we see them as strong evidence that the proposed methodology could benefit other more sophisticated approaches, as it is orthogonal to most other works in the domain.
>
> > Missing ablation: The authors note that they choose the top 50 classes (AA) in the informed settings to create the subsets, mainly to keep ‘similar size of subsets’ to the uninformed settings; [...]
>
> We added an experiment varying the value of M (the size of the class subset) to see its influence on performance. Results are presented in the Appendix in Figure 2. It can be seen that any value of M between roughly 20 and 80 can lead consistently to improved results on most datasets, showing robustness in the selection of the optimum M.
> We agree that this value is likely dependent on the base dataset. We observed that clusters of about 50 classes obtained in the UI setting were typically quite coherent. It is not necessarily expected that such clusters would be obtainable on any base dataset.
> We also showcased the value of M obtained by using a cumulative total (activation) mass reaching a proportion of 90%, to illustrate that this method would indeed lead to a good selection in our setting.
>
> > [...] the random subsets seem to be surprisingly better when using AA, FIM and RH – [...] why this might be the case?
>
> We currently do not have an unequivocal explanation. We are investigating this and we think it might be related to the fact that choosing between feature extractors specialized on meaningful subsets (X) can actually lead to worse outcomes than some feature extractors generalist subsets (R).
>
> > [...] Semantic features generally provide better ‘upper bounds’ [...]; Do the authors have some intuition whether that’s due to the ‘more powerful’ underlying CLIP training, or whether semantics (text) generally provide richer features for such selection? [...]
>
> This is a very interesting phenomenon. We believe the two hypotheses you propose to be relevant. We think one way to answer this question would be to compare the visual and semantic features of CLIP (which allegedly have features of equal “representational power” ).
>
> > [...] However, doesn’t MCS actually have quite high compute requirements? [...]
>
> We changed the text to better reflect what we meant. In short, the UI setting is particularly interesting for cases where latency is constrained, as it does not require finetuning a specific backbone. Yet, among all heuristics, MCS is more costly than others, and as such there is open room for improvements in the heuristics for future work.

---

> ### Comment · Reviewer_snfD · 2023-11-22
> **Thanks for the detailed response**
>
> I would like to thank the authors for their detailed response and additional information.
> While I do think this work provides interesting insights that are somewhat orthogonal to most other works, I do share the concerns that it is difficult to directly generalize/make use of the findings beyond the investigated settings in a robust manner.
> $\rightarrow$ I will hence stick with my initial rating.

---

> > ### Author Response · Authors · 2023-11-23
> >
> > Thanks for your response. While we accept the possibility that certain choices in the implementation of the approach may need to be adapted for different datasets or modalities, we argue that the extensive evaluation in the paper (8 few-shot datasets, 3 task samplers and 3 novel settings) provides strong evidence for the fundamental idea that fine-tuning with a subset of base classes can be an effective mechanism to tailor features to a few-shot task. We believe that this is valuable knowledge for the few-shot learning community. To help us improve the paper in the future, we would be grateful if you could provide suggestions of how to expand the investigation to provide greater confidence in the generality of the result.

---

### Official Review · Reviewer_oGJc · 2023-11-03

**Soundness:** 2 fair
**Presentation:** 2 fair
**Contribution:** 2 fair
**Rating:** 5
**Confidence:** 4

**Summary:**

This paper studies the few-shot transfering by fine-tuning on a subset of base classes, proposes simple strategies to select base classes under multiple few-shot scenarios. Better performances are achieved compared to the baseline without fine-tuning.

**Strengths:**

The paper provides a thorough evaluation of various downstream tasks under multiple setups.

The paper is well-written.

**Weaknesses:**

The paper, while presenting a method for selecting base classes and achieving better performance with fine-tuning in few-shot scenarios, lacks a strong sense of novelty. It would be beneficial if the authors could establish the optimality of their method, specifically by showing that it can identify the best 50 classes from a pool of 1000 ImageNet classes. Alternatively, if reaching the optimum is unfeasible, the paper should strive to provide approximate solutions that approach the upper bound.

Furthermore, while the paper does show improvements over a baseline without fine-tuning, it would be more insightful if the upper bound, representing the optimal 50 classes for each task, could be explored and discussed, if practical. This would provide a clearer perspective on the significance of the achieved results.

**Questions:**

What is the optimal solution? And how to approach it?

---

> ### Author Response · Authors · 2023-11-21
>
> We would like to thank the reviewer for their comments. In the following, we address their concerns and questions to the best of our ability.
>
> > It would be beneficial if the authors could establish the optimality of their method, specifically by showing that it can identify the best 50 classes from a pool of 1000 ImageNet classes. Alternatively, if reaching the optimum is unfeasible, the paper should strive to provide approximate solutions that approach the upper bound.
>
> > it would be more insightful if the upper bound, representing the optimal 50 classes for each task, could be explored and discussed
>
> > What is the optimal solution? And how to approach it?
>
>
> > The paper, while presenting a method for selecting base classes and achieving better performance with fine-tuning in few-shot scenarios, lacks a strong sense of novelty. It would be beneficial if the authors could establish the optimality of their method, specifically by showing that it can identify the best 50 classes from a pool of 1000 ImageNet classes. Alternatively, if reaching the optimum is unfeasible, the paper should strive to provide approximate solutions that approach the upper bound.
> Furthermore, while the paper does show improvements over a baseline without fine-tuning, it would be more insightful if the upper bound, representing the optimal 50 classes for each task, could be explored and discussed, if practical. This would provide a clearer perspective on the significance of the achieved results.
>
> We respectfully disagree with the reviewer about the novelty of our approach. To our knowledge, we are the first to clearly hypothesize, and then empirically demonstrate, that fine-tuning to a subset of the base classes can actually result in improved overall accuracy. This finding opens a new line of research, as most papers do not question the base dataset and use it as a whole, indivisible entity.
>
> We sincerely would like to provide results with optimality, but we believe that this is a very challenging question, probably beyond our reach. From a combinatorial perspective, there are 712 chooses 50 possible subsets to create, from which we know no theoretical tool that can predict the performance.
>
> Heuristics, such as greedy selection of the best classes for a target domain, could be added to the paper, but this would inevitably look arbitrary and lacking depth. We think that this question should be considered as future work, keeping our main contribution a demonstration of the ability of improving performance by selecting a subset of base classes.
>
> The approach we introduced and discussed in the paper significantly improves over the baseline, showing that we can improve on the performance in a very competitive field. We think that this should be weighted positively by the reviewer.

---

### Official Review · Reviewer_fEBb · 2023-11-06

**Soundness:** 3 good
**Presentation:** 3 good
**Contribution:** 2 fair
**Rating:** 6
**Confidence:** 3

**Summary:**

- This paper examines the effectiveness of using a pre-trained feature extractor on a smaller, related subset of data to improve few-shot learning, where traditional fine-tuning on very small target datasets is often ineffective.

- The study explores different few-shot learning scenarios across eight domains, showing that selecting a subset of base classes closer to the target dataset's distribution can enhance performance.

- The authors propose simple, intuitive methods for few-shot classification improvement, providing insights into when these methods are most effective and releasing their experimental code on GitHub for reproducibility.

**Strengths:**

- The paper explores an approach where a pre-trained model, also referred to as a "base model" or "feature extractor," is fine-tuned using a carefully selected subset of the classes from its original training dataset—the "base dataset". This selection process involves choosing classes that are most relevant to the task the model will perform after fine-tuning. The selected classes should be diverse enough to prevent the fine-tuning process from causing the model to overfit to a small, non-representative sample of data. At the same time, it narrows down the scope of learning to what's most pertinent for the target application.

**Weaknesses:**

- There is no investigation into exactly why or how fine-tuning on fewer classes helps. The theoretical understanding is limited.

- The simplicity of the NCM classifier limits what conclusions can be drawn about the quality of representations. Testing on more complex classifiers would strengthen the results.

- Clustering classes for the static library has no guarantee of generating coherent subgroups. Better ways to determine class groupings could be developed rather than simple hierarchical clustering.

**Questions:**

- The class centroids used for NCM classification could be skewed by the fine-tuning. Could improvements just come from this distortion rather than better representations?

- Can the authors provide a rigorous statistical or empirical rationale for selecting $M=50$ as the fixed number of base classes in the Average Activations (AA) and Unbalanced Optimal Transport (UOT) selection strategies? How does this choice influence the balance between the breadth of class representation and the manageability of the subset size, particularly in relation to the diversity of the domain examples $D$ and the overall size of the base class set $\mathcal{C}$? Additionally, what impact might this have on the representational capacity of the fine-tuned feature extractor, especially when considering domains with a significantly higher or lower intrinsic class cardinality?

- How do the proposed heuristics for selecting specialist feature extractors address the issue of distributional shifts between the labeled support set and the unseen query set in few-shot learning tasks? Considering that these heuristics (SSA, SSC, LOO, SNR, RKM, MCS, FIM, AA) rely on the assumption that the support set is a representative sample of the task's data distribution, isn't there a significant risk that the heuristics will fail to predict feature extractor performance accurately in the presence of such shifts?

- How can the method be robustified to account for potential discrepancies between the support and query distributions, which are common in real-world scenarios?

---

> ### Author Response · Authors · 2023-11-21
>
> We would like to thank the reviewer for their constructive comments and interesting questions. In the following, we address their concerns and questions to the best of our ability.
>
> >  [...] The theoretical understanding is limited.
>
> We rephrased the motivation in the main paper. In brief, the question is not really about fine-tuning, but rather about the selection of the base dataset for a given target task. There seems to be an implicit belief in the community that the larger the base dataset is, the better will be the backbone, no matter the task. Recent contributions using foundation models as the backbone are going even further down this path. We wrote this contribution because we wanted to question this belief. Our starting point is that there exists a large number of subset of classes, and that some of them may actually help improve performance on downstream tasks, in particular when these downstream tasks are specialised to a subdomain of the base dataset. The main question addressed in our paper is thus: if we increase the number of possible backbones to choose from by artificially creating subsets of base classes, can we find a compromise between having enough of them so that some are going to improve the performance on our tasks, and at the same time few enough so that we can select one that is improving performance using the very limited number of data points on our task?
>
> Nonetheless, we recognise the value of a theoretical framework in this area and hope that future research will explore this issue.
>
> > [...] Testing on more complex classifiers would strengthen the results.
>
> We agree that NCM is limited as a representative of existing strategies. We have added experiments using logistic regression. Results (in Table 2), where we see very similar conclusions to the case of NCM. We would like to thank the reviewer for pointing out this concern as it helped broaden our claims.
>
>
> > Better ways to determine class groupings could be developed [...]
>
> We agree with the reviewer that clustering might not be the best approach here. However, since the scope of the paper is mainly to show the existence of strategies that can improve performance, we did not want to dive too much into the optimization of these strategies. (see Appendix for more detail).
>
> > [...] Could improvements just come from this distortion rather than better representations?
>
> It is a very interesting question. To investigate whether the improvement is due to mere distortion as opposed to a better representation, we conducted a set of experiments where the backbone was frozen and only an additional linear layer was trained on the class subset. This linear layer can thus distort the feature space without fundamentally changing the representation. Our results (Table 6) show that our method is more effective than a mere distortion of the original representation.
>
>
> >Can the authors provide a rigorous statistical or empirical rationale for selecting M=50 [...]
>
> That is indeed an important question that we spent time investigating during the rebuttal. We ran experiments to observe the effect of varying M in the top-M class selection. (as shown in Figure 2 of the Appendix). We see that the accuracy change is not particularly sensitive to M given M is sufficiently large. As a rule of thumb, any M above 20 classes and below 80 classes is a good fit for every sampling and datasets, justifying the choice of 50. These numbers should be considered highly dependent on the fact we are using ImageNet, on which there exists coherent subsets of classes with dozens of elements each, as displayed in the Appendix. Considering a different base dataset, we expect that the choice of M can be challenging, or require input from an expert.
>
>
> > How do the proposed heuristics for selecting specialist feature extractors address the issue of distributional shifts between the labeled support set and the unseen query set in few-shot learning tasks? [...] How can the method be robustified [...]?
>
> We agree that this is an interesting and important question within few-shot learning, as addressed in recent works such as Bennequin et al., “Bridging Few-Shot Learning and Adaptation: New Challenges of Support-Query Shift”. However, for the purposes of this investigation, we have restricted our scope to the simpler but common setting where the support set and query set can be assumed to be drawn from the same distribution. We have added a few lines in Section 4 acknowledging this limitation and citing a relevant work.
>
> That said, we believe there is potential to investigate a transductive variant of our algorithm that is informed by the query set as well as the support set or domain. Techniques such as transported prototypes (Bennequin et al.) could be used to seek an alignment of the distributions. Alternatively, in the UI setting, one could incorporate the distance between the two distributions as part of a heuristic for selecting a feature transform from the static library.

---

### Official Review · Reviewer_u4xN · 2023-11-08

**Soundness:** 2 fair
**Presentation:** 2 fair
**Contribution:** 2 fair
**Rating:** 5
**Confidence:** 4

**Summary:**

This paper empirically investigate how to learn better features for the target dataset by training on fewer base classes. Authors propose several heuristics for selecting a feature extractor. Extensive experiments in eight domains demonstrate the effectiveness of the proposed heuristics.

**Strengths:**

1. The writing is clear and easy to follow.

2. Compared to the baselines, there are consistent improvements, especially with domain-informed settings.

**Weaknesses:**

1. In general, authors investigate a collection of heuristics to select subsets for training. While some of these heuristics are effective in certain domains, it is difficult to draw a concise conclusion (e.g., Figure 1). It is more like empirical analysis than a novel approach.

2. In Table 1, the performance improvements seem to dependent on the similarities between the classes of target datasets and those of base classes (e.g., Aircraft and Traffic Signs). This may limit the application of the proposed heuristics.

**Questions:**

See the weakness

---

> ### Author Response · Authors · 2023-11-21
>
> We would like to thank the reviewer for their perspective. In the following, we address their concerns and questions to the best of our ability.
>
> > In general, authors investigate a collection of heuristics to select subsets for training. While some of these heuristics are effective in certain domains, it is difficult to draw a concise conclusion (e.g., Figure 1). It is more like empirical analysis than a novel approach.
>
> We respectfully disagree with the reviewer. We would like to point out that the collection of heuristics that is mentioned in the comment is not our main contribution in this paper. As stated in the paper, the main contribution is to show that there exist subsets of base classes that, when fine-tuned on, can lead to improved performance in few-shot classification of images, and that such subsets can be retrieved even when very limited knowledge of the target task is available. As far as we know, this is indeed a novel approach, and it offers new opportunities for improving few-shot classification. It is true that the paper is mostly empirical, but we still believe it can constitute a significant contribution to the field due to the originality of the described approach.
>
> > In Table 1, the performance improvements seem to dependent on the similarities between the classes of target datasets and those of base classes (e.g., Aircraft and Traffic Signs). This may limit the application of the proposed heuristics.
>
> In fact, while class similarity is one important factor, our experiments on a wide variety of domains indicate that it is not the *only* factor determining the success of the method. For example, while the Fungi and VGG Flower domains have several similar base classes in ImageNet, and the Omniglot domain has none, we observe larger improvements for Omniglot using our method in the DI and TI settings (see Table 1 and pie charts in appendix). This shows that the simple Average Activation strategy is capable of identifying a subset of useful classes even when the distributions of the base classes and target classes do not overlap at all.

---

### Official Review · Reviewer_8Z55 · 2023-11-09

**Soundness:** 3 good
**Presentation:** 3 good
**Contribution:** 4 excellent
**Rating:** 6
**Confidence:** 4

**Summary:**

This paper investigates few-shot learning. They propose a novel pipeline that fine-tunes a pre-trained model on a subset of training examples ("base classes"), which are similar to the testing domain, to emphasize these parts and benefit the testing performance. They investigate three settings of tasks, including informed and un-informed, and design specific methods for them. The experiment results show that the proposed pipeline significantly improves the results.

**Strengths:**

- The proposed idea of emphasizing the most relevant subset as the testing domain in the pre-training dataset and the corresponding pipeline is novel.
- The proposed method is simple and effective. The authors cover three reasonable settings and provide reasonable methods for them.
- The experiment results show that the improvement is significant.
- The idea of emphasizing a subset of the pre-training dataset is novel and instructive direction in meta-learning research.

**Weaknesses:**

- Some necessary contents, including Alg. 1 and details of heuristics, are put in the appendix. This makes the audience unable to see a rigid, complete version of their method from a main paper, raising concern about page limit circumvention.
- In Sec. 3.2 about informed settings, a method is directly put there without any analysis, reasons, and details. I would like the authors to provide the motivation and some details behind these methods: Why is AA designed in this form? What is the implementation of $g$, fine-tuning on a pre-trained, fixed $h$; using a non-learning method on $g$; or co-train $h$ and $g$ on the pre-trained dataset? Is the same $g$ used in the latter testing phase? Does the scale of $g$ (as logits) matter in selecting $\mathcal{X}$?
- In Sec. 3.3 about uninformed settings, the authors introduce the method just by listing and citing a number of related works. So, I would like to know what the authors contribute to this part, or whether they are just summarizing or aggregating existing methods.
- All the tasks are image classifications. I wonder if other tasks, other than image classification, can be applied with the proposed pipeline. I would like to see some simple yet representative results on any other tasks. If the proposed method can be applied not only in image classification but also in many other tasks, the contribution of this paper can be even higher.
- (Minor) In Alg. 1, the set of examples is defined as $\mathcal{X}$, but the pseudo code uses $X$. And, line 2 is actually not required since operation 3 "selecting top $M$" already requires a top-$M$ partition.

**Questions:**

In "Weaknesses".

If the authors could address some of my concerns and provide results on some tasks other than image classification, I will raise the score.

---

> ### Author Response · Authors · 2023-11-21
>
> We would like to thank the reviewer for their positive and constructive comments. In the following, we address their concerns and questions to the best of our ability.
>
> > Some necessary contents, including Alg. 1 and details of heuristics, are put in the appendix. This makes the audience unable to see a rigid, complete version of their method from a main paper, raising concern about page limit circumvention.
>
> We made some minor changes to free enough space so that Alg. 1 is now part of the main paper.
>
> > In Sec. 3.2 about informed settings, a method is directly put there without any analysis, reasons, and details. I would like the authors to provide the motivation and some details behind these methods: Why is AA designed in this form? What is the implementation of g, fine-tuning on a pre-trained, fixed ℎ; using a non-learning method on h; or co-train ℎ and g on the pre-trained dataset? Is the same used in the latter testing phase? Does the scale of g (as logits) matter in selecting X ?
>
> We have revised the description of the method and moved the algorithm into the main paper. See revised paragraph below:
>
> > To choose a class subset, we need a method by which to identify the base classes which are most useful for a given this set of examples $\mathcal{X}$. Fortunately, the base model already comprises a classifier that assigns a score to each base class. We therefore propose to simply compute the average class likelihoods predicted by the base model on the novel set~$\mathcal{X}$, and then select the $M$ highest-scoring classes. This straightforward selection strategy will henceforth be referred to as Average Activations (AA), and is outlined in Algorithm 1. While it is by no means guaranteed to select the class subset that will yield the optimal representation for the final task after fine-tuning, it is a reasonable proxy for that purpose.
>
> We hope this makes clear that the $g$ used for AA subset selection is precisely that which was learned with the base feature transform $h$, without additional fine-tuning. Fine-tuning is only applied *after* class subset selection, to adapt the base feature transform to the selected subset.
>
>
> > In Sec. 3.3 about uninformed settings, the authors introduce the method just by listing and citing a number of related works. So, I would like to know what the authors contribute to this part, or whether they are just summarizing or aggregating existing methods.
>
> Section 3.3 describes our procedure for building a static library of feature extractors in the uninformed setting, for which we adopt Ward’s linkage algorithm and CLIP text embeddings. Indeed, the contribution here lies not in any novel algorithm for clustering or text embedding, but rather in the idea of applying these standard tools to build such a library of feature extractors by fine-tuning on class subsets. We designed the algorithm to be as simple and straightforward as possible since our purpose was not to over-optimize the feature extractors for the Meta-Dataset domains, but rather to demonstrate that the approach can lead to consistent improvements in accuracy even when not tailored to a specific target dataset. We clarified the text.
>
> > If the proposed method can be applied not only in image classification but also in many other tasks, the contribution of this paper can be even higher.
>
> We would like to thank the reviewer for the interesting suggestion. We designed a new experiment where the DI feature extractors are used to perform semantic segmentation afterwards, using only one image for each considered class. We used Cityscape and adapted the backbones to these tasks by adding a small multilayer subnetworks trained on the few available shots, a methodology similar to Mining latent classes for few-shot segmentation by Lihe Yang et al.  As can be seen in Table 5 in the Appendix, we obtained results consistent with classification with significant increase in performance in the DI setting.
>
> We also fixed the typo you found in the algorithm.

---

### Author Response · Authors · 2023-11-23

At the conclusion of the discussion phase, we wish to thank all reviewers for their work and highlight the key improvements made in light of their comments.
* We conducted experiments using logistic regression as the few-shot learner rather than NCM, and obtained a consistent improvement relative to logistic regression using the generic features. This demonstrates that our findings are not specific to the choice of few-shot learner.
* We performed fine-tuning with different subset sizes $M$ for all 8 domains and produced a plot. This revealed that almost all domains have an optimal subset size which is much smaller than the full base dataset. It also showed that the choice of $M = 50$ was well justified.
* We trained feature extractors from scratch instead of fine-tuning in the Domain-Informed setting, and this resulted in lower few-shot classification accuracy for all domains except Omniglot. This could be due to the relative lack of data or the hyper-parameters being optimized for the full dataset, however it remains true that fine-tuning on a subset of the data yields a better representation.
* We tested whether the improvement was due to a mere distortion of the feature space (rather than a better representation) by training only a linear layer on top of the frozen baseline features. This resulted in consistently worse accuracy even when varying the dimension of the projection, showing that the fine-tuning does not simply distort the existing feature space.
* To extend the findings beyond image classification, we applied the approach to few-shot semantic segmentation with the Cityscapes dataset. We were able to achieve an improvement in accuracy and mIOU.

All results have been included in the updated PDF. We also made several edits, including moving Algorithm 1 into the main text for completeness.

Note: Besides the main revision of the PDF uploaded on 21/11, we made a minor revision on 23/11 with improved results for few-shot segmentation

---

### Meta-Review · Area_Chair_x85L · 2023-12-12

**Metareview:**

Paper Summary - The paper investigates how to select a relevant subset of base classes for fine-tuning a pre-trained representation to improve few-shot learning on novel target classes.  This is achieved by constructing three settings of increasing difficulty, namely domain-informed, task-informed, and uninformed, where information about the target task progressively decreases. Several heuristics are introduced for the selection of relevant base classes. The experiments present results on eight target domains from MetaDataset in the cross-domain setting. The findings demonstrate that through fine-tuning the feature extractor on a careful selection of base classes, enhanced performance can be achieved for most target domains. This improvement is validated when using a nearest class mean classifier (or logistic regression classifier, as clarified in the rebuttal), surpassing baselines without fine-tuning and fine-tuning solely on novel classes. The rebuttal also includes results for few-shot semantic segmentation on the Cityscapes dataset.

Strengths and Weaknesses - While the reviewers recognize that the paper presents an interesting idea for underscoring the most relevant subset of base classes in few-shot learning, with a straightforward yet effective method that leads to noticeable performance improvements across several target tasks, they also have various concerns regarding the original submission and the revised paper.

First, a major issue is the unclear practical implications and broader applicability of the findings for the few-shot learning community beyond the specific settings investigated in this paper. Although the idea of selecting relevant base classes that are potentially similar to target novel classes seems intuitively helpful, its feasibility in practice might be questionable. This is particularly true when considering factors such as proprietary base classes, the availability of only pre-trained models (i.e., foundation models), or a substantial domain gap between base and novel classes. It is unclear if the study conducted in this paper would inspire crucial directions for future research in few-shot learning. Meanwhile, although the study in this paper differs somewhat from most other works, the evaluation primarily centers on comparisons with naive baselines. The generalizability of the proposed approach remains uncertain, particularly in its ability to enhance state-of-the-art few-shot learning methods that utilize more sophisticated strategies to adapt pre-trained representations. It would be of broader impact if such an evaluation had been investigated in the paper as well.

Also, as noted by the reviewers, the proposed base class selection strategies, along with the chosen number of base classes, seem somewhat heuristic. In the rebuttal, the authors presented additional results with varying numbers of base classes. The reviewers suggested that introducing a more principled selection mechanism and providing an in-depth discussion on optimality analysis would considerably improve the paper's quality.

**Justification For Why Not Higher Score:**

The authors' rebuttal clarified some confusion and provided additional experimental results, such as using logistic regression classifiers and extending the study to few-shot semantic segmentation tasks. However, the major concerns mentioned above were not adequately addressed. Therefore, I cannot recommend acceptance based on the current version.

**Justification For Why Not Lower Score:**

N/A

---

### Decision · Program_Chairs · 2024-01-16

Reject